# GAN "Steerability" Without Optimization

**Nurit Spingarn Eliezer**$^{\circ\dagger}$          **Ron Banner**$^{\dagger}$          **Tomer Michaeli**$^{\circ}$

Technion – Israel Institute of Technology$^{\circ}$ , Habana Labs - Intel $^{\dagger}$

{nurits@campus,tomer.m@ee}.technion.ac.il, ron.banner@intel.com

## Abstract

Recent research has shown remarkable success in revealing "steering" directions in the latent spaces of pre-trained GANs. These directions correspond to semantically meaningful image transformations (*e.g.*, shift, zoom, color manipulations), and have similar interpretable effects across all categories that the GAN can generate. Some methods focus on user-specified transformations, while others discover transformations in an unsupervised manner. However, all existing techniques rely on an optimization procedure to expose those directions, and offer no control over the degree of allowed interaction between different transformations. In this paper, we show that "steering" trajectories can be computed in *closed form* directly from the generator's weights without any form of training or optimization. This applies to user-prescribed geometric transformations, as well as to unsupervised discovery of more complex effects. Our approach allows determining both linear and non-linear trajectories, and has many advantages over previous methods. In particular, we can control whether one transformation is allowed to come on the expense of another (*e.g.*, zoom-in with or without allowing translation to keep the object centered). Moreover, we can determine the natural end-point of the trajectory, which corresponds to the largest extent to which a transformation can be applied without incurring degradation. Finally, we show how transferring attributes between images can be achieved without optimization, even across different categories.

## 1 Introduction

Since their introduction by Goodfellow et al. (2014), generative adversarial networks (GANs) have seen remarkable progress, with current models capable of generating samples of very high quality (Brock et al., 2018; Karras et al., 2019a; 2018; 2019b). In recent years, particular effort has been invested in constructing controllable models, which allow manipulating attributes of the generated images. These range from disentangled models for controlling *e.g.*, the hair color or gender of facial images (Karras et al., 2019a;b; Choi et al., 2018), to models that even allow specifying object relations (Ashual & Wolf, 2019). Most recently, it has been demonstrated that GANs trained without explicitly enforcing disentanglement, can also be easily "steered" (Jahanian et al., 2020; Plumerault et al., 2020). These methods can determine semantically meaningful linear directions in the latent space of a pre-trained GAN, which correspond to various different image transformations, such as zoom, horizontal/vertical shift, in-plane rotation, brightness, redness, blueness, etc. Interestingly, a walk in the revealed directions typically has a similar effect across all object categories that the GAN can generate, from animals to man-made objects.

To detect such latent-space directions, the methods of Jahanian et al. (2020) and Plumerault et al. (2020) require a training procedure that limits them to transformations for which synthetic images can be produced for supervision (*e.g.*, shift or zoom). Other works have recently presented unsupervised techniques for exposing meaningful directions (Voynov & Babenko, 2020; Härkönen et al., 2020; Peebles et al., 2020). These methods can go beyond simple user-specified transformations, but also require optimization or training of some sort (*e.g.*, drawing random samples in latent space).

In this paper, we show that for most popular generator architectures, it is possible to determine meaningful latent space trajectories directly from the generator's weights without performing any kind of training or optimization. As illustrated in Fig. 1, our approach supports both simple *user-defined geometric transformations*, such as shift and zoom, and *unsupervised exploration* of directions that typically reveals more complex controls, like the 3D pose of the camera or the blur of the

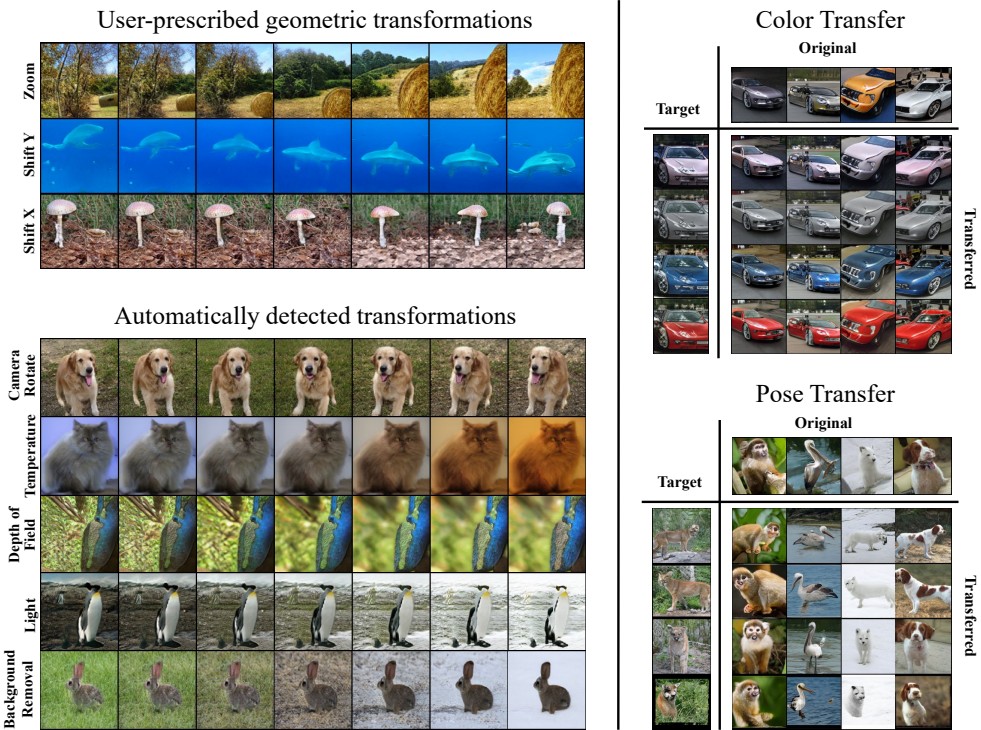

Figure 1: **Steerability without optimization.** We determine meaningful trajectories in the latent space of a pre-trained GAN without using optimization. We accommodate both user-prescribed geometric transformations, and automatic detection of semantic directions. We also achieve attribute transfer without any training. All images were generated with BigGAN (Brock et al., 2018).

background. We also discuss how to achieve attribute transfer between images, even across object categories (see Fig. 1), again without any training. We illustrate results mainly on BigGAN, which is class-conditional, but our trajectories are class-agnostic. Our approach is advantageous over existing methods in several respects. First, it is $10^4 \times$-$10^5 \times$ faster. Second, it seems to detect more semantic directions than other methods. And third, it allows explicitly accounting for dataset biases.

**First order dataset biases**     As pointed out by Jahanian et al. (2020), dataset biases affect the extent to which a pre-trained generator can accommodate different transformations. For example, if all objects in the training set are centered, then no walk in latent space typically allows shifting an object too much without incurring degradation. This implies that a "steering" latent-space trajectory should have an end-point. Our nonlinear trajectories indeed possess such convergence points, which correspond to the maximally-transformed versions of the images at the beginning of the trajectories. Conveniently, the end-point can be computed in closed form, so that we can directly jump to the maximally-transformed image without performing a gradual walk.

**Second order dataset biases**     Dataset biases can also lead to coupling between transformations. For example, in many datasets zoomed-out objects can appear anywhere within the image, while zoomed-in objects are always centered. In this case, trying to apply a zoom transformation may also result in an undesired shift so as to center the enlarged object. Our unsupervised method allows controlling the extent to which transformation A comes on the expense of transformation B.

## 1.1    RELATED WORK

**Walks in latent space**    Many works use walks in a GAN's latent space to achieve various effects (*e.g.*, (Shen et al., 2020; Radford et al., 2015; Karras et al., 2018; 2019b; Denton et al., 2019; Xiao et al., 2018; Goetschalckx et al., 2019)). The recent works of Jahanian et al. (2020) and Plumerault et al. (2020) specifically focus on determining trajectories which lead to simple user-specified transformations, by employing optimization through the (pre-trained) generator. Voynov & Babenko

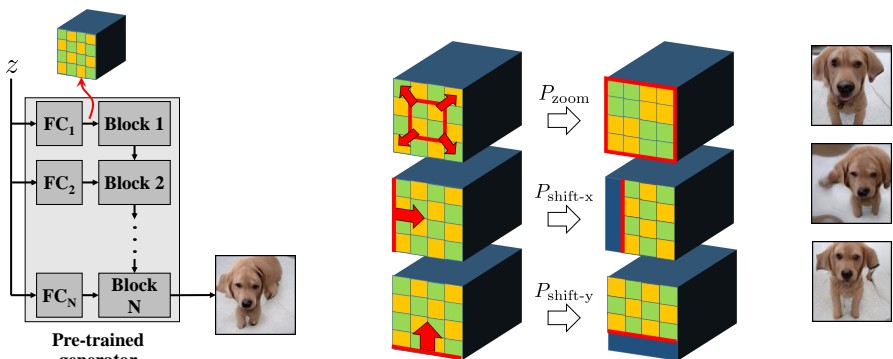

Figure 2: **User-prescribed spatial manipulations.** We calculate directions in latent space whose effect on the tensor at the output of the first layer, is similar to applying transformation $\boldsymbol{P}$ on that tensor. This results in the generated image experiencing the same transformation.

(2020) proposed an unsupervised approach for revealing dominant directions in latent space. This technique reveals more complex transformations, such as background blur and background removal, yet it also relies on optimization. Most recently, the work of Härkönen et al. (2020) studied unsupervised discovery of meaningful directions by using PCA on deep features of the generator. The method seeks linear directions in latent space that best map to those deep PCA vectors, and results in a set of non-orthogonal directions. Similarly to the other methods, it also requires a very demanding training procedure (drawing random latent codes and regressing the latent directions), which can take a day for models like BigGAN.

**Nonlinear walks in latent space**   Linear latent-space trajectories may arrive at regions where the probability density is low. To avoid this, some methods proposed to replace the popular Gaussian latent space distribution by other priors (Kilcher et al., 2018), or to optimize the generator together with the latent space (Bojanowski et al., 2018). Others suggested to use nonlinear walks in latent space that avoid low-probability regions. For example, Jahanian et al. (2020) explored nonlinear trajectories parametrized by two-layer neural networks, while White (2016) proposed spherical paths for interpolating between two latent codes.

**Hierarchical GAN architectures**   Recently there is tendency towards hierarchical GAN architectures (Karras et al., 2018; 2019a; Brock et al., 2018; Choi et al., 2018), which are capable of producing high resolution images at very high quality. It is known that the earlier scales in such models are responsible for generating the global composition of the image, while the deeper scales are responsible for more local attributes (Karras et al., 2019a; Yang et al., 2019; Härkönen et al., 2020). Here, we distil this common knowledge and show how meaningful directions can be detected in each level, and how these architectures allow transferring attributes between images.

## 2 USER-SPECIFIED GEOMETRIC TRANSFORMATIONS

Most modern generator architectures map a latent code vector $\boldsymbol{z} \in \mathbb{R}^d$ having no notion of spatial coordinates, into a two-dimensional output image. In some cases (*e.g.*, BigGAN), different parts of $\boldsymbol{z}$ are processed differently. In others (*e.g.*, BigGAN-deep), $\boldsymbol{z}$ is processed as a whole. However, in all cases, the first layer maps $\boldsymbol{z}$ (or part of it) into a tensor with low spatial resolution (*e.g.*, $4 \times 4 \times 1536$ in BigGAN 128). This tensor is then processed by a sequence of convolutional layers that gradually increase its spatial resolution (using fractional strides), until reaching the final image dimensions.

Our key observation is that since the output of the first layer already has spatial coordinates, this layer has an important role in determining the coarse structure of the generated image. This suggests that if we were to apply a geometric transformation, like zoom or shift, on the output of the first layer, then we would obtain a similar effect to applying it directly on the generated image (Fig. 2). In fact, it may even allow slight semantic changes to take place due to the deeper layers that follow, which can compensate for the inability of the generator to generate the precise desired transformed image. As we now show, this observation can be used to find latent space directions corresponding to simple geometric transformations.

## 2.1 LINEAR TRAJECTORIES

Let us start with linear trajectories. Given a pre-trained generator $G$ and some transformation $\mathcal{T}$, our goal is to find a direction $q$ in latent space such that $G(z + q) \approx \mathcal{T}\{G(z)\}$ for every $z$. To this end, we define $P$ to be the matrix corresponding to $\mathcal{T}$ in the resolution of the first layer's output. Denoting the weights and biases of the first layer by $W$ and $b$, respectively, our goal is therefore to bring[1] $W(z + q) + b$ as close as possible to $P(Wz + b)$. To guarantee that this holds *on average* over random draws of $z$, we formulate our problem as

$$\min_{q} \; \mathbb{E}_{z \sim p_z} \left[ \left\| D\Big( W(z + q) + b - P(Wz + b) \Big) \right\|^2 \right], \tag{1}$$

where $p_z$ is the probability density function of $z$, and $D$ is a diagonal matrix that can be used to assign different weights to different elements of the tensors. For example, if $P$ corresponds to a horizontal shift of one element to the right, then we would not like to penalize for differences in the leftmost column of the shifted feature maps (see Fig. 2). In this case, we set the corresponding diagonal elements of $D$ to $0$ and the rest to $1$. Assuming $\mathbb{E}[z] = 0$, as is the case in most frameworks, the objective in (1) simplifies to

$$\mathbb{E}_{z \sim p_z} \left[ \left\| D\Big( (I - P)Wz \Big) \right\|^2 \right] + \left\| D\Big( Wq + (I - P)b \Big) \right\|^2, \tag{2}$$

where $I$ is the identity matrix. The first term in (2) is independent of $q$, and the second term is quadratic in $q$ and is minimized by

$$q = \left( W^T D^2 W \right)^{-1} W^T D^2 (P - I)\, b. \tag{3}$$

We have thus obtained a closed form expression for the optimal linear direction corresponding to transformation $P$ in terms of only the weights $W$ and $b$ of the first layer.

Figure 2 illustrates this framework in the context of the BigGAN model, in which the feature maps at the output of the first layer are $4 \times 4$. For translation, we use a matrix $P$ that shifts the tensor by one element (aiming at translating the output image by one fourth its size). For zoom-in, we use a matrix $P$ that performs nearest-neighbor $2\times$ up-sampling, and for zoom-out we use sub-sampling by $2\times$. For each such transformation, we can control the extent of the effect by multiplying the steering vector $q$ by some $\alpha > 0$.

Figure 1 (top-left) and Fig. 3(a) show example results for zoom and shift with the BigGAN generator. As can be seen, this simple approach manages to produce pronounced effects, although not using optimization through the generator, as in (Jahanian et al., 2020). Following (Jahanian et al., 2020), we use an object detector to quantify our zoom and shift transformations. Figure 4 shows the distributions of areas and centers of object bounding boxes in the transformed images. As can be seen, our trajectories lead to similar effects to those of Jahanian et al. (2020), despite being $10^4 \times$ faster to compute (see Tab. 1). Please refer to App. A.1 for details about the evaluation, and see additional results with BigGAN and with the DCGAN architecture of (Miyato et al., 2018) in App. A.3.

## 2.2 ACCOUNTING FOR FIRST-ORDER DATASET BIASES VIA NEUMANN TRAJECTORIES

With linear trajectories, the generated image inevitably becomes improbable after many steps, as $p_z(z + \alpha q)$ is necessarily small for large $\alpha$. This causes the generated image to distort until eventually becoming meaningless after many steps. One way to remedy this, is by using nonlinear trajectories that have endpoints. Here, we focus on walks in latent space, having the form

$$z_{n+1} = M z_n + q, \tag{4}$$

for some matrix $M$ and vector $q$. We coin these Neumann trajectories, since unfolding the iterations leads to a Neumann series. An important feature of such walks is that if the spectral norm of $M$ is strictly smaller than $1$ (a condition we find to be satisfied in practice for the optimal $M$), then they have a convergence point. We use a diagonal $M$, which we find gives the best results. To determine the optimal $M$ and $q$ for a transformation $P$, we modify Problem (1) into

$$\min_{M, q} \; \mathbb{E}_{z \sim p_z} \left[ \left\| D\Big( W(Mz + q) + b - P(Wz + b) \Big) \right\|^2 \right]. \tag{5}$$

---

[1] For architectures like BigGAN, in which the first FC layer operates on a *subset* of the entries of the latent vector, we use $z$ to refer to this subset rather than to the whole vector.

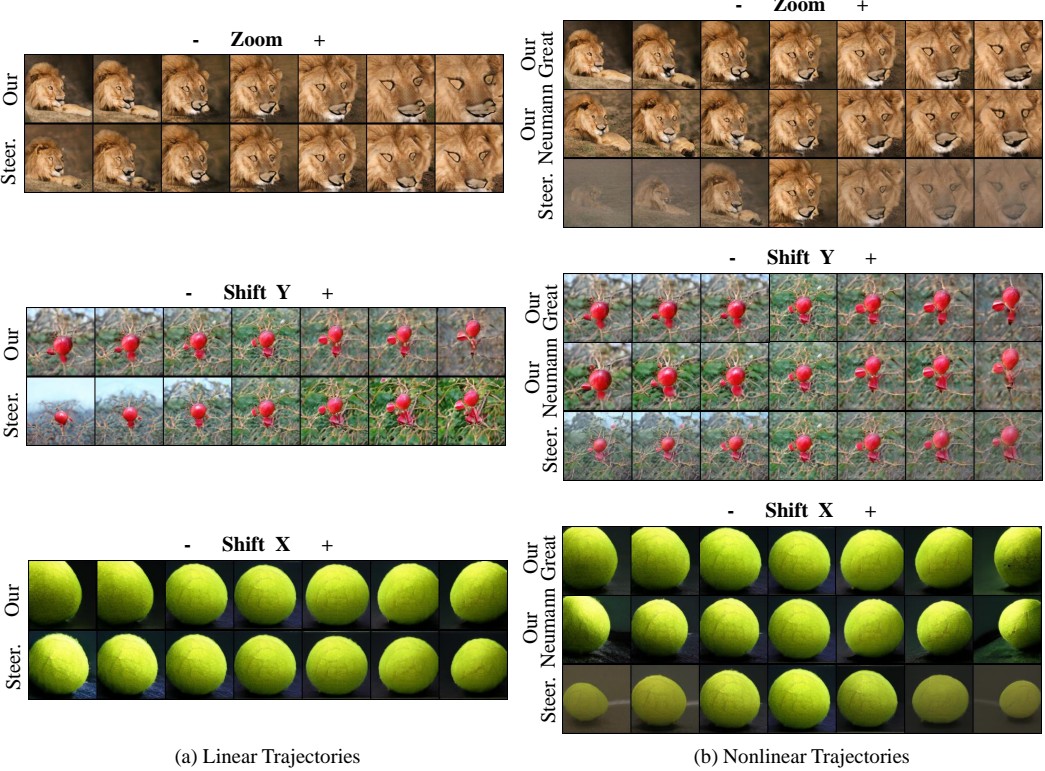

(a) Linear Trajectories         (b) Nonlinear Trajectories

Figure 3: **Walks corresponding to geometric transformations.** We compare our zoom and shift trajectories to those of the GAN steerability work (Jahanian et al., 2020). For linear paths, the methods are qualitatively similar, whereas for nonlinear walks, our methods are advantageous.

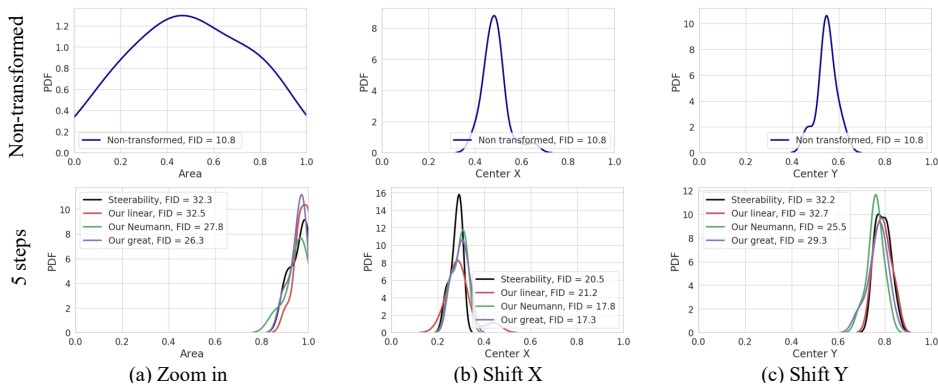

(a) Zoom in         (b) Shift X         (c) Shift Y

Figure 4: **Quantitative comparison with (Jahanian et al., 2020).** We show the probability densities of object areas and locations after 2 (top) and 5 (bottom) steps of walks for BigGAN-128. The step-size is the same for the linear walks, and matches the size of the first step of the nonlinear walk. Our walks have similar effects to those of Jahanian et al. (2020), with the nonlinear variants achieving lower FID scores after 5 steps, at the cost of only slightly weaker transformation effects.

We assume again that $\mathbb{E}[\boldsymbol{z}] = 0$, and make the additional assumption that $\mathbb{E}[\boldsymbol{z}\boldsymbol{z}^T] = \sigma_z^2 \boldsymbol{I}$, which is the case in all current GAN frameworks. In this setting, the objective in (5) reduces to

$$\sigma_z^2 \left\| \boldsymbol{D}\Big(\boldsymbol{W}\boldsymbol{M} - \boldsymbol{P}\boldsymbol{W}\Big) \right\|_{\mathrm{F}}^2 + \left\| \boldsymbol{D}\Big(\boldsymbol{W}\boldsymbol{q} + (\boldsymbol{I} - \boldsymbol{P})\boldsymbol{b}\Big) \right\|^2, \tag{6}$$

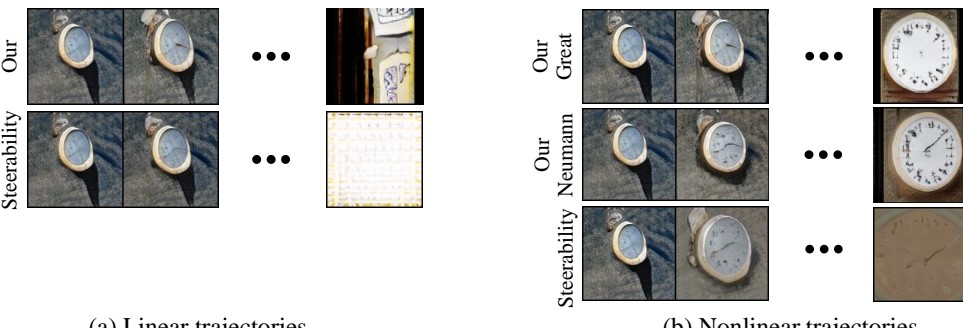

(a) Linear trajectories          (b) Nonlinear trajectories

Figure 5: **Endpoints.** (a) Linear walks eventually lead to deteriorated images (shown here for zoom). (b) Our nonlinear walks converge to meaningful images. The nonlinear trajectories of the GAN steerability method (Jahanian et al., 2020) also converge, but always to the same (unnatural) image for a given class.

where $\|\cdot\|_F$ denotes the Frobenius norm. Here, $q$ appears only in the second term, which is identical to the second term of (2). Therefore, the optimal $q$ is as in (3). The matrix $M$ appears only in the first term, which is easily shown to be minimized when setting the diagonal entries of $M$ to

$$M_{i,i} = \frac{w_i^T D^2 P\, w_i}{w_i^T D^2\, w_i}, \tag{7}$$

where $w_i$ is the $i$th column of $W$.

**Controlling the step size**    As opposed to linear trajectories, refining the step size along our curved trajectories necessitates modifying both $M$ and $q$. To do so, we can search for a matrix $\tilde{M}$ and vector $\tilde{q}$ with which $N$ steps of the form $z_{n+1} = \tilde{M} z_n + \tilde{q}$ are equivalent to a single step of the walk (4). Noting that the $N$th step of the refined walk can be explicitly written as $z_N = \tilde{M}^N z_0 + \left(\sum_{k=0}^{N-1} \tilde{M}^k\right)\tilde{q}$, we conclude that the parameters of this $N$-times finer walk are

$$\tilde{M} = M^{\frac{1}{N}}, \qquad \tilde{q} = \left(\sum_{k=0}^{N-1} M^{\frac{k}{N}}\right)^{-1} q. \tag{8}$$

**Convergence point**    If the spectral norm of $M$ is smaller than 1, then we have that

$$\lim_{n\to\infty} z_n = \lim_{n\to\infty}\left(M^n z_0 + \left(\sum_{k=0}^{n-1} M^k\right) q\right) = (I - M)^{-1} q, \tag{9}$$

where we used the fact that the first term tends to zero and the second term is a Newmann series. Superficially, this may seem to imply that the endpoint of the trajectory is not a function of the initial point $z_0$. However, recall that in hierarchical architectures, like BigGAN, $z$ refers to the part of the latent vector that enters the first layer. The rest of the latent vector is not modified throughout the walk. Therefore, the latent vector at the endpoint equals the latent vector of the initial point, except for its subset of entries corresponding to the first hierarchy level, which are replaced by $(I-M)^{-1} q$.

### 2.3 ACCOUNTING FOR FIRST-ORDER DATASET BIASES VIA GREAT CIRCLE TRAJECTORIES

In the Neumann walk, the step size decreases along the path (as $\|z_{n+1} - z_n\| \to 0$). We now discuss an alternative nonlinear trajectory that has a natural endpoint yet permits a constant step size. Here we avoid low density regions by explicitly requiring that the likelihood of all images along the path is constant. For $z \sim \mathcal{N}(0, I)$, this translates to the requirement that the whole trajectory lie on the sphere whose radius equals the norm of the original latent code $z_0$. We stress that the method we discuss here can be applied to any direction $q$, whether determined in a supervised manner or not.

Specifically, suppose we want to steer our latent code towards a normalized direction $v = q /\|q\|$. Then we can walk along the great circle on the sphere that passes through our initial point $z_0$, and

the point $\|\boldsymbol{z}_0\|\boldsymbol{v}$ (blue circle in Fig. 6). Mathematically, let $\mathcal{V}$ denote the (one-dimensional) subspace spanned by $\boldsymbol{v}$ and let $\boldsymbol{P}_{\mathcal{V}} = \boldsymbol{v}\boldsymbol{v}^T$ and $\boldsymbol{P}_{\mathcal{V}^\perp} = \boldsymbol{I} - \boldsymbol{P}_{\mathcal{V}}$ denote the orthogonal projections onto $\mathcal{V}$ and $\mathcal{V}^\perp$, respectively. Then the great circle trajectory can be expressed as

$$\boldsymbol{z}_n = \|\boldsymbol{z}_0\| \left( \boldsymbol{u}\cos(n\Delta + \theta) + \boldsymbol{v}\sin(n\Delta + \theta) \right), \tag{10}$$

where $\boldsymbol{u} = \boldsymbol{P}_{\mathcal{V}^\perp}\boldsymbol{z}_0/\|\boldsymbol{P}_{\mathcal{V}^\perp}\boldsymbol{z}_0\|$ and $\theta = \arccos(\boldsymbol{P}_{\mathcal{V}^\perp}\boldsymbol{z}_0/\|\boldsymbol{z}_0\|) \times \mathrm{sign}(\langle \boldsymbol{z}_0, \boldsymbol{v}\rangle)$. The effect of this trajectory for a zoom-in direction is shown in Fig. 6 (third row). The natural endpoint of the great-circle path is $\|\boldsymbol{z}_0\|\boldsymbol{v}$ (blue point), beyond which the contribution of $\boldsymbol{v}$ starts to decrease. As seen in Fig. 6, this endpoint indeed corresponds to a plausible zoomed-in version of the original image.

## 2.4 Comparison

Figure 3(b) compares our nonlinear walks (Neumann and great-circle) with those of the GAN steer-abilty work of Jahanian et al. (2020). As can be seen, the latter tend to involve undesired brightness changes. The advantage of our nonlinear trajectories over the linear ones becomes apparent when performing long walks, as exemplified in Fig. 5. In such settings, the linear trajectories deteriorate, whereas our nonlinear paths have meaningful endpoints. This can also be seen in Fig. 4, which reports the Frećhet Inception distances (FID) achieved by the two approaches. Interestingly, the nonlinear trajectories of the GAN steerability method also have endpoints, but these endpoints are the same for all images of a certain class (and distorted).

## 3 Unsupervised exploration of transformations

To go beyond simple user-prescribed geometric transformations, we now discuss exploration of additional manipulations in an unsupervised manner. The key feature of our approach is that by revealing a large set of directions, we can now also account for second-order dataset biases.

## 3.1 Principal latent space directions

We start by seeking a set of orthonormal directions (possibly a different set for each generator hierarchy) that lead to the maximal change at the output of the layer to which $\boldsymbol{z}$ is injected. These directions are precisely the right singular vectors of the corresponding weight matrix $\boldsymbol{W}$, i.e., the $k$th most significant direction is the $k$th column of the matrix $\boldsymbol{V}$ in the singular value decomposition $\boldsymbol{W} = \boldsymbol{U}\boldsymbol{S}\boldsymbol{V}^T$ (assuming the diagonal entries of $\boldsymbol{S}$ are arranged in decreasing order). This reveals directions corresponding to many geometric, texture, color, and background effects (see Fig. 1).

Our approach is seemingly similar to GANspace (Härkönen et al., 2020), which computes PCA of activations within the network. However, they optimize over latent space directions that best map to this deep PCA basis. Concretely, they feed-forward random latent codes $\{\boldsymbol{z}^{(j)}\}$ to obtain deep-feature representations $\{\boldsymbol{y}^{(j)}\}$, compute the PCA

| Method | Memory | Time |
|---|---|---|
| Jahanian et al. (2020) | 0 | 40 min (per dir.) |
| Härkönen et al. (2020) | 1GB | 14 hrs (all) |
| Voynov & Babenko (2020) | 0 | 10 hrs (all) |
| Our principal directions | 0 | **327 ms** (all) |

Table 1: Complexity for BigGAN-deep-512.

basis $\boldsymbol{A}$ and mean vector $\boldsymbol{\mu}$ of these features, and then solve for a steering basis $\boldsymbol{V} = \arg\min \sum_j \|\boldsymbol{V}\boldsymbol{A}^T(\boldsymbol{y}^{(j)} - \boldsymbol{\mu}) - \boldsymbol{z}^{(j)}\|$. Thus, besides computational inefficiency (see Tab. 1), they obtain a set of non-orthogonal latent-space directions (see App. Fig. 48) that correspond to repeated effects (see App. Figs.42-46). In contrast, our directions are orthogonal by construction, and therefore capture a more diverse set of effects (see App. Figs.42-46). For example, the semantic dis-similarity between $G(\boldsymbol{z})$ and $G(z + 3\boldsymbol{v})$ is $64\%$ larger with our method, as measured by the average LPIPS distance (Zhang et al., 2018) over the first 50 directions ($33 \cdot 10^{-3}$ for GANSpace, $54 \cdot 10^{-3}$ for us).

Having determined a set of semantic directions, we now want to construct trajectories that exhibit the corresponding effects, but also account for dataset biases. As discussed in Sec. 2 and illustrated in the first two rows of Fig. 6, performing linear walks along these directions eventually leads to distorted images. A more appropriate choice is thus to use the great-circle walk described in Sec. 2.

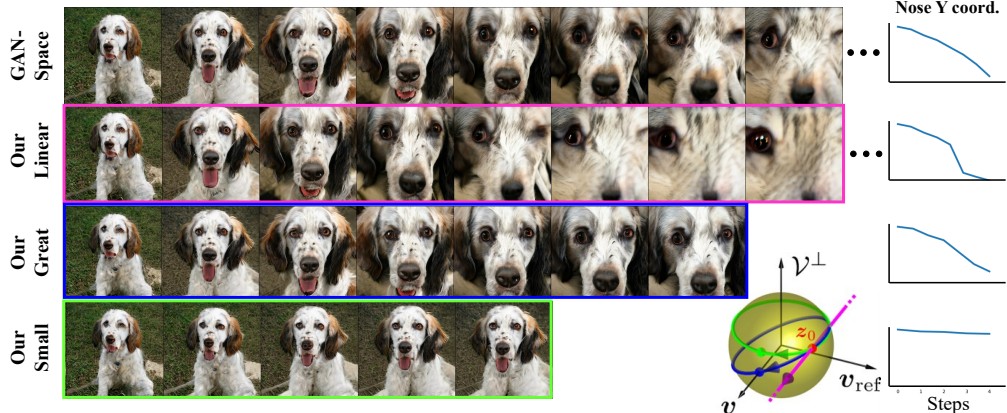

Figure 6: **Orbits in latent space.** A linear trajectory (magenta) in the principal direction $v$ corresponding to zoom, eventually draws apart from the sphere and results in distorted images. The great circle (blue) that connects $z_0$ with $\|z_0\|v$ keeps the image natural all the way, but allows also other transformations (shift in this case). The small circle (green) that only modifies $v_{\text{ref}}$ in addition to $v$, does not induce any other transformation besides zoom ($v_{\text{ref}}$ is the least dominant direction). Particularly, it keeps the nose's vertical coordinate fixed (right plots). See also App. Figs. 40-41

This is illustrated in the third row of Fig. 6. While leading to meaningful endpoints, a limitation of the great circle trajectory is that when walking on the sphere towards $v$, we actually also modify the projections onto other principal directions. This causes other properties to change besides the desired attribute. For example, in Fig. 6, the great circle causes a shift, centering the dog in addition to the principal zoom effect (see the nose position graphs on the right). This stems from a second-order dataset bias. Indeed, as shown in Fig. 7, BigGAN generates small (zoomed-out) dogs at almost any location within the image, but its generated large (zoomed-in) dogs tend to be centered.

## 3.2 ACCOUNTING FOR SECOND-ORDER DATASET BIASES VIA SMALL CIRCLE TRAJECTORIES

Using our set of directions to battle second-order biases is non-trivial, as walking on the sphere towards $v$ while keeping the projections onto all other principal directions fixed is impossible (it induces too many constraints). However, we note that if we allow the projection onto only one of the other directions, say $v_{\text{ref}}$, to change, then it becomes possible to keep the projections onto all other axes fixed. Such a trajectory is in fact a small circle on the sphere, that lies in the affine subspace that contains $z_0$ and is parallel to $\mathcal{V} = \text{span}\{v, v_{\text{ref}}\}$. Specifically, the small circle walk is given by

$$z_n = P_{\mathcal{V}^\perp} z_0 + \|P_{\mathcal{V}} z_0\|(v_{\text{ref}} \cos(n\Delta + \theta) + v \sin(n\Delta + \theta)), \tag{11}$$

where $\theta = \arccos(P_{\mathcal{V}_{\text{ref}}} z_0 / \|P_{\mathcal{V}} z_0\|) \times \text{sign}(\langle P_{\mathcal{V}} z_0, v \rangle)$ with $P_{\mathcal{V}_{\text{ref}}} = v_{\text{ref}} v_{\text{ref}}^T$. One natural choice for $v_{\text{ref}}$ is the principal direction having the smallest singular value, which corresponds to the weakest effect. As can be seen in the bottom row of Fig. 6, the small circle trajectory with this choice leads to a zoom effect without shift or any other dominant transformation. This is also illustrated in Fig. 7, which shows the distribution of the horizontal translation between the initial point and the endpoint of the trajectory. As can be seen, the small circle walk incurs the smallest shift and keeps the FID highest, albeit leading to a slightly smaller zoom effect. In App. A.2 we show additional examples, including with different choices of $v_{\text{ref}}$.

## 4 ATTRIBUTE TRANSFER

In the previous sections we explicitly computed directions in latent space. An alternative way of achieving a desired effect, is to transfer attributes from a different image. As we now show, this can also be achieved without optimization. Specifically, in App. A.2 we show that for BigGAN, principal directions corresponding to different hierarchies control distinctively different attributes. Now, our key observation is that this allows transferring attributes between images, simply by copying from a target image the part of $z$ corresponding to a particular hierarchy (see Fig. 8). For example, to transfer *pose*, we replace the part corresponding to the first level. As seen in Figs. 1 and 8, this al-

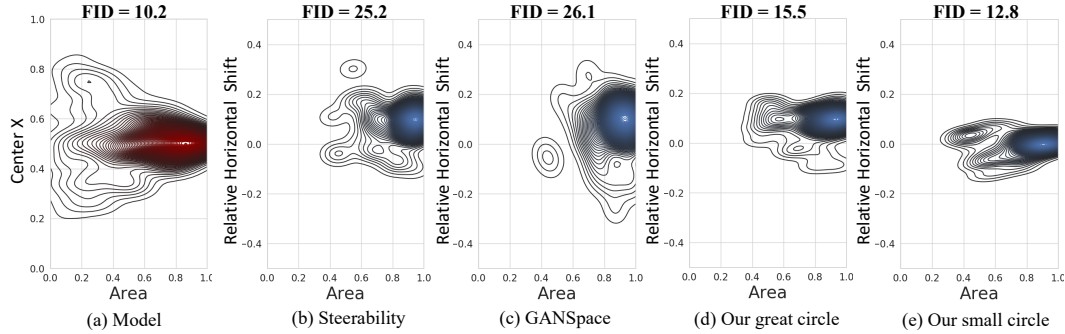

(a) Model     (b) Steerability     (c) GANSpace     (d) Our great circle     (e) Our small circle

Figure 7: **Accounting for second-order dataset bias.** In red is the joint distribution of area and horizontal center of BigGAN-generated Labrador dogs. This plot shows that zoomed-out dogs can appear anywhere, whereas zoomed-in dogs are mostly centered. In blue are the joint distributions of area and horizontal translation (namely delta shift) achieved by walks in a zoom-in direction. All walks indeed increase the area, but also undesirably shift the dog. Our methods incur smaller shifts, with the small circle walk incurring negligible shift. From left the right, the mean shifts of the methods are 0.08, 0.10, 0.06 and 0.01. This allows us to achieve lower FIDs, but at the cost of achieving slightly smaller zoom effects (the mean areas are 0.85, 0.83, 0.80 and 0.76).

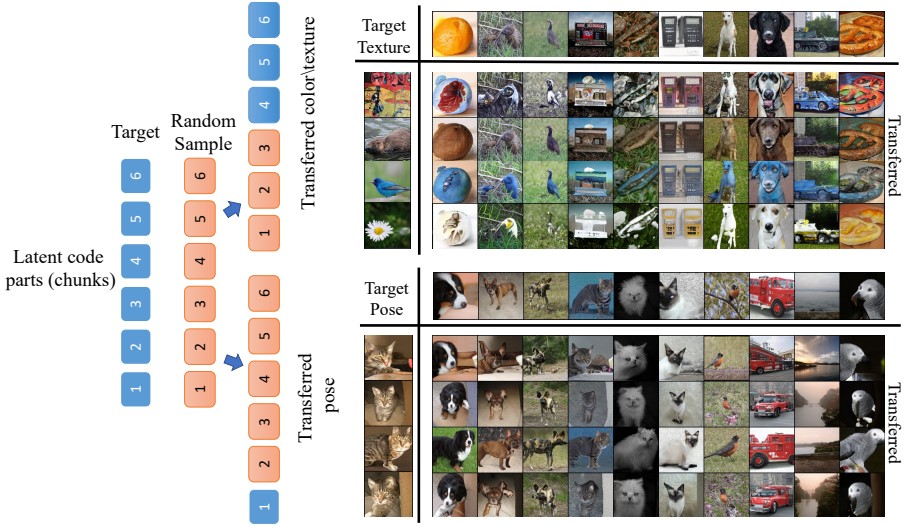

Figure 8: **Attributes transfer.** In BigGAN-128 the latent code is divided into 6 chunks that are injected to different hierarchy levels. Transferring pose, color or texture, can be done by copying specific parts of the latent code from the target image.

lows transferring pose even across classes. Within the same class, we can transfer *color* by copying the elements of hierarchies 4,5 and 6 and *texture* by copying hierarchies 3,4 and 5 (see Appendix for more examples). Note that unlike other works discussing semantic style hierarchies (*e.g.*, (Karras et al., 2019a; Yang et al., 2019)), our pre-trained BigGAN was not trained to disentangle attributes.

## 5 CONCLUSION

We presented methods for determining paths in the latent spaces of pre-trained GANs, which correspond to semantically meaningful transformations. Our approach extracts those trajectories directly from the generator's weights, without requiring optimization or training of any sort. Our methods are significantly more efficient than existing techniques, they determine a larger set of distinctive semantic directions, and are the first to allow explicitly accounting for dataset biases.

**Acknowledgements** This research was supported by the Israel Science Foundation (grant 852/17) and by the Technion Ollendorff Minerva Center.

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

## A GAN "STEERABILITY" WITHOUT OPTIMIZATION: APPENDIX

### A.1 QUANTITATIVE EVALUATION

We adopt the method proposed in Jahanian et al. (2020) and utilize the *MobileNet-SSD-V1* detector[2] to estimate object bounding boxes. To quantify shifts, we extract the centers of the bounding boxes along the corresponding axis. To quantify zoom, we use the area of the bonding boxes. In the following paragraphs, we elaborate on all quantitative evaluations reported in the main text.

**Figure 4** Here, we show the probability densities of object areas and locations after 2 (top) and 5 (bottom) steps. Since we use unit-norm direction vectors, the length of the linear paths we walk through are 2 and 5 as well. As for the nonlinear path, we choose the first step to have the same length. However, the overall length of the path is different. For example, on average, five steps of the nonlinear trajectory have a total length of $5.95$, but reach at a point of distance only $4.3$ from the initial point. We used 100 randomly chosen classes from the ImageNet dataset, and 30k images from each class. The same images are used for both the FID measurement and for generating the PDF plots.

**Figure 6** In order to ensure that each step of the linear walk and the great and small circle walks has the same geodesic distance, we set

$$\Delta_L = \Delta_G \|\boldsymbol{z}_0\| = \Delta_S \|\boldsymbol{P}_{\mathcal{V}} \boldsymbol{z}_0\|, \tag{12}$$

where $\Delta_L$, $\Delta_G$ and $\Delta_S$ are the step sizes of the linear, great circle and small circle walks, respectively. This ensures that the arc-length of a step on the circles is the same as the length of a step of the linear walk.

**Figure 7** Here, we aim to demonstrate a particular second order dataset bias. We chose 10 classes which we found to exhibit strong coupling between the size and location of the object. For example, dogs, cats and in general, animals. We plotted 80 levels-sets of 2D KDEs computed using the seaborne package. In Fig. 7 we show results for a Labrador retriever dog, we observed similar results for the classes: golden retriever (207), Welsh springer spaniel (218), Great grey Owl (24), Persian cat (283), plane (726), tiger (292), Old English sheepdog (229), passenger car (705), goose (99), husky (248). See Figs. 40 and 41 for additional results.

**Table 1** In Tab. 1, we compare the running time and memory usage of all methods. For[3] (Jahanian et al., 2020), we measure the time it takes to learn one direction, which includes the training process. For[4] (Härkönen et al., 2020), we measure the total time it takes to extract the directions, including the sample collection, the PCA, and the regression. We noticed that the regression stage was the heaviest. As for our method, we measure the time it takes the CPU to perform SVD. The column "Memory" specifies the required memory for collecting samples. Only GANSpace (Härkönen et al., 2020) requires that stage.

### A.2 UNSUPERVISED EXPLORATION OF PRINCIPAL DIRECTIONS

#### A.2.1 COMPARISONS WITH RANDOM DIRECTIONS

In Fig. 10 - 9 we explore principal directions via linear walks, using the same initial image (in the middle). In Figs. 14-16 we explore the transformations that arise in each hierarchy of BigGAN-128. Specifically, we compare our linear directions which are based on SVD, with random directions. We draw 5 different directions from an isotropic Gaussian distribution and normalize them to have unit-norms, similarly to our directions. Then, we linearly add them to the initial latent code with fixed number of steps. We can observe that each random direction induces a different complex effect, which cannot be described by a single semantic property. For examples, in the first random direction (R1) we can see rotation, zoom and background changes, while in the third (R3), there is a kind of vertical shift. On the other hand, our principal directions show one prominent transformation

---

[2]https://github.com/qfgaohao/pytorch-ssd
[3]https://github.com/ali-design/
[4]https://github.com/harskish/ganspace/

for each scale. We focus on directions that have the same effect for all classes and do not show directions that lead to different effects for different classes, like changes of day-night in one class and background in another class.

### A.2.2 ALTERNATIVE SMALL CIRCLE WALKS

In Figs. 29-33, we show more examples, this time with small circle walks towards principal directions. In all those examples, the reference direction $v_{ref}$ for the small circle, is the least dominant direction (namely, the singular vector with the smallest singular value). This ensures that when walking towards the principal direction $v$, we modify no other dominant property. That is, we modify the property associated with $v$ without modifying the properties associated with any other principal direction, besides $v_{ref}$ (which is the least dominant one). In Figs. 29-33 we show some cases in which the initial generated image is not in the middle of the small circle path and therefore in these cases, we need to take a different number of steps to each side. The endpoints are defined as the points where the cosine in Eq. 11 becomes 0 and 1.

We do not have to choose the reference direction $v_{ref}$ to be the least dominant one. If we choose it to be a dominant direction, then we may obtain various interesting phenomena, depending on the interaction between the directions $v$ and $v_{ref}$. This is illustrated in Figs. 37-39. Specifically, in Fig. 37 and 38, we perform a walk in the direction corresponding to zoom, while allowing only the vertical shift to change. In this case, the walk manages to center the object so as to achieve a significant zoom effect. In Fig. 39, on the other hand, we perform a walk in the direction corresponding to zoom while allowing only the rotation to change. Here, the zoom effect is less dominant, but we do see a strong rotation effect.

### A.2.3 SECOND ORDER DATASET BIASES

In Figs. 40 and 41 we show more examples for second order dataset biases. Specifically, those figures depict the joint distributions of area and horizontal center shift (top) and area and vertical center shift (bottom) at the end of walks that are supposed to induce only zoom-in. Our small circle walks exhibit the smallest undesired shifts.

### A.2.4 COMPARISONS WITH GANSPACE

In Fig. 42-46, we show visual comparisons with GANSpace (Härkönen et al., 2020). We specifically focus on the first 50 directions founded by each method and show that our linear directions lead to stronger effects for most of the directions. All directions were scaled to have a unit norm and are linearly added or subtracted from the initial latent code with the same step size. In Fig. 48, we show that our direction are orthogonal to each other much more then the directions found by (Härkönen et al., 2020).

### A.3 USER PRESCRIBED SPATIAL MANIPULATIONS

We provide additional examples for walks corresponding to user prescribed geometric transformations. We focus on zoom, vertical shift and horizontal shift, and show both linear and our nonlinear trajectories.

### A.3.1 COMPARISONS WITH JAHANIAN ET AL.

In Figs. 49 we show additional comparisons with Jahanian et al. (2020).

### A.3.2 ADDITIONAL RESULTS

In Figs. 50-53 we show additional zoom trajectories and and in Fig. 54, 55 additional shift trajectories. As can be seen, the linear trajectories often remain more loyal to the original image (at the center) after a small number of steps. However, for a large number of steps, the nonlinear trajectories lead to more plausible images.

### A.3.3    RESULTS ON DCGAN

In Figs. 56 - 58, we show results with ResNet based GAN presented in Miyato et al. (2018). That GAN has a FC layer as the first stage, which is all we need in order to perform our spatial manipulations and to extract principal components. Since that architecture is not an hierarchical one, we can manipulate the first layer only.

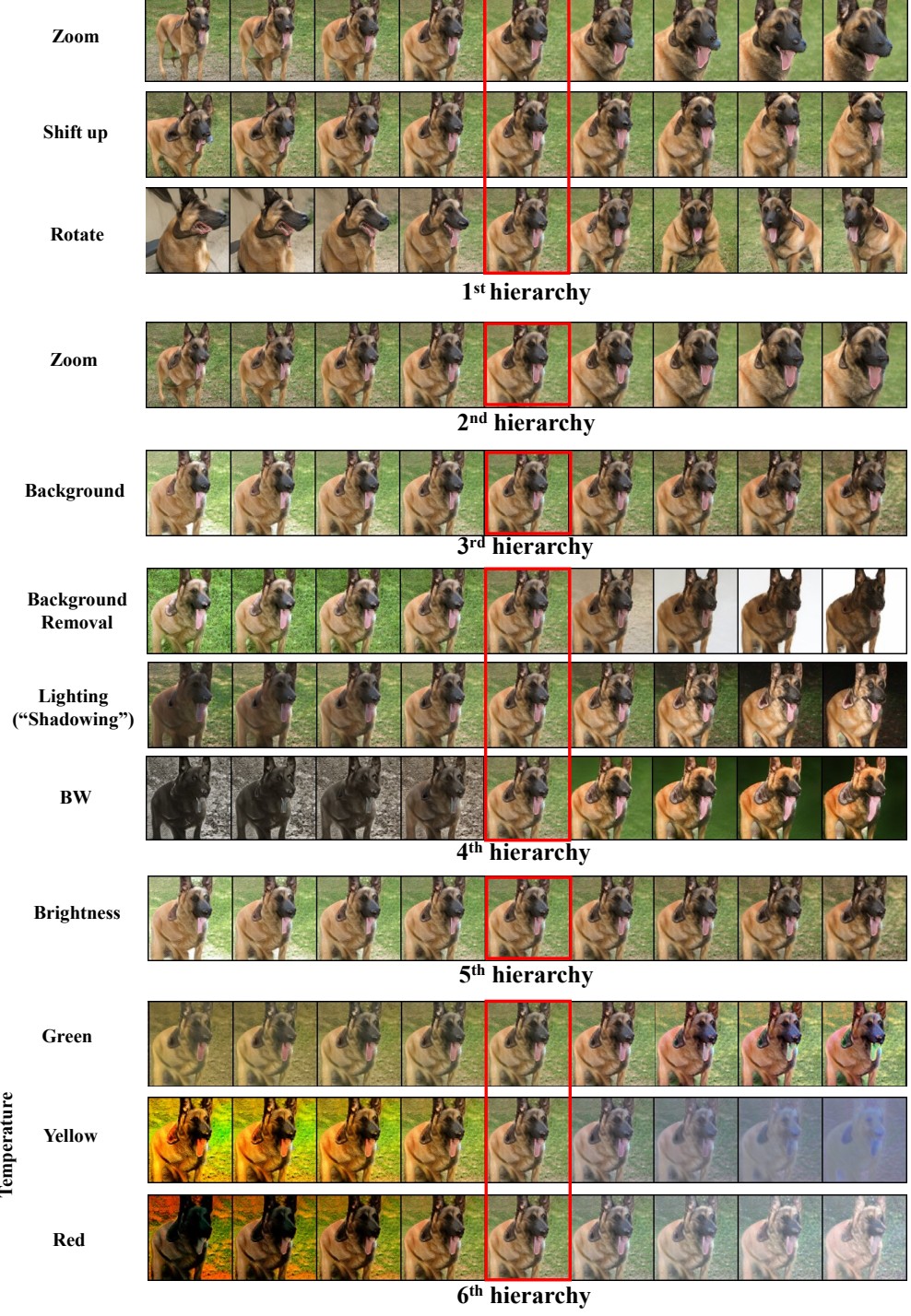

Figure 9: Our explored directions in BigGAN.

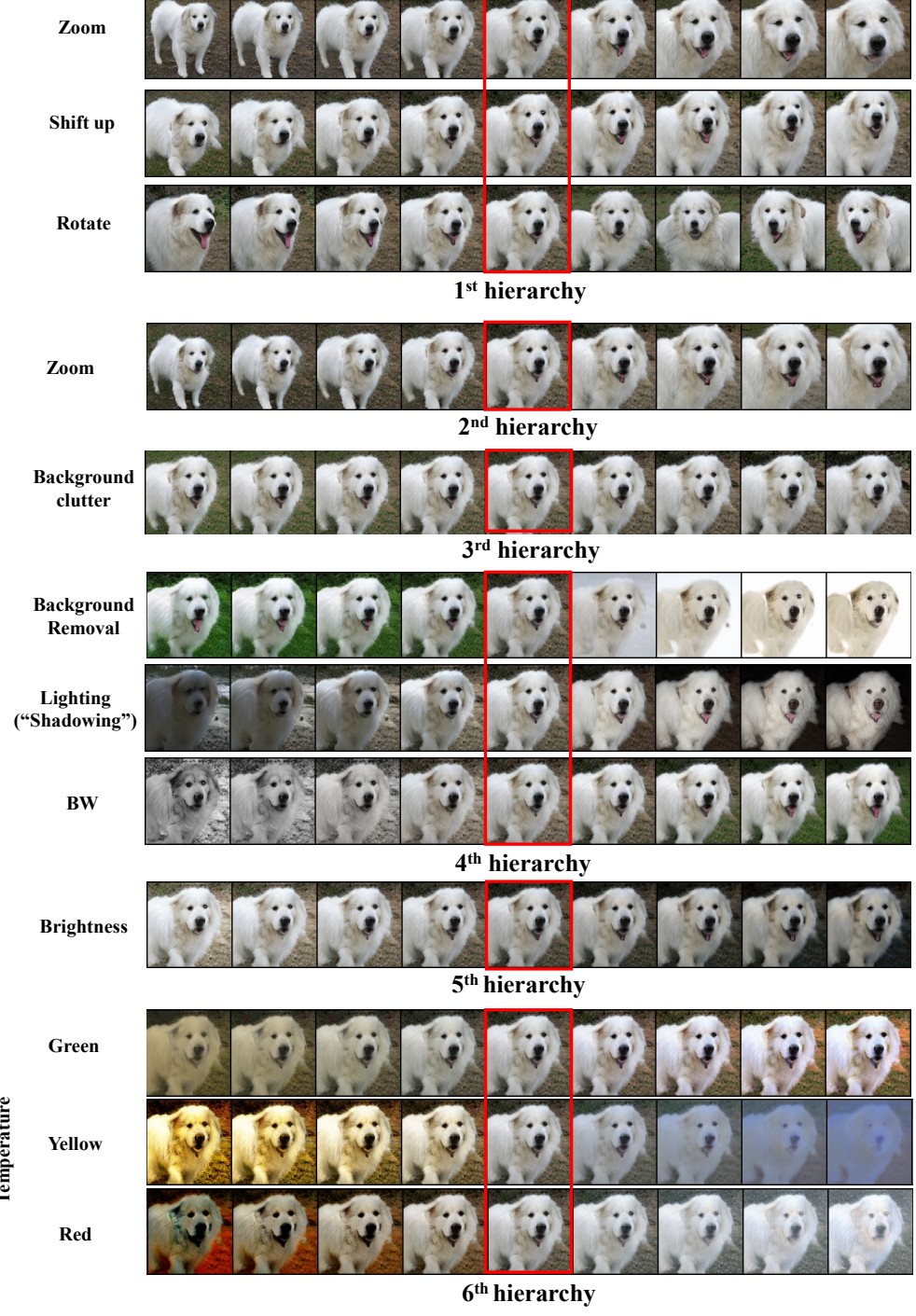

Figure 10: Our explored directions in BigGAN.

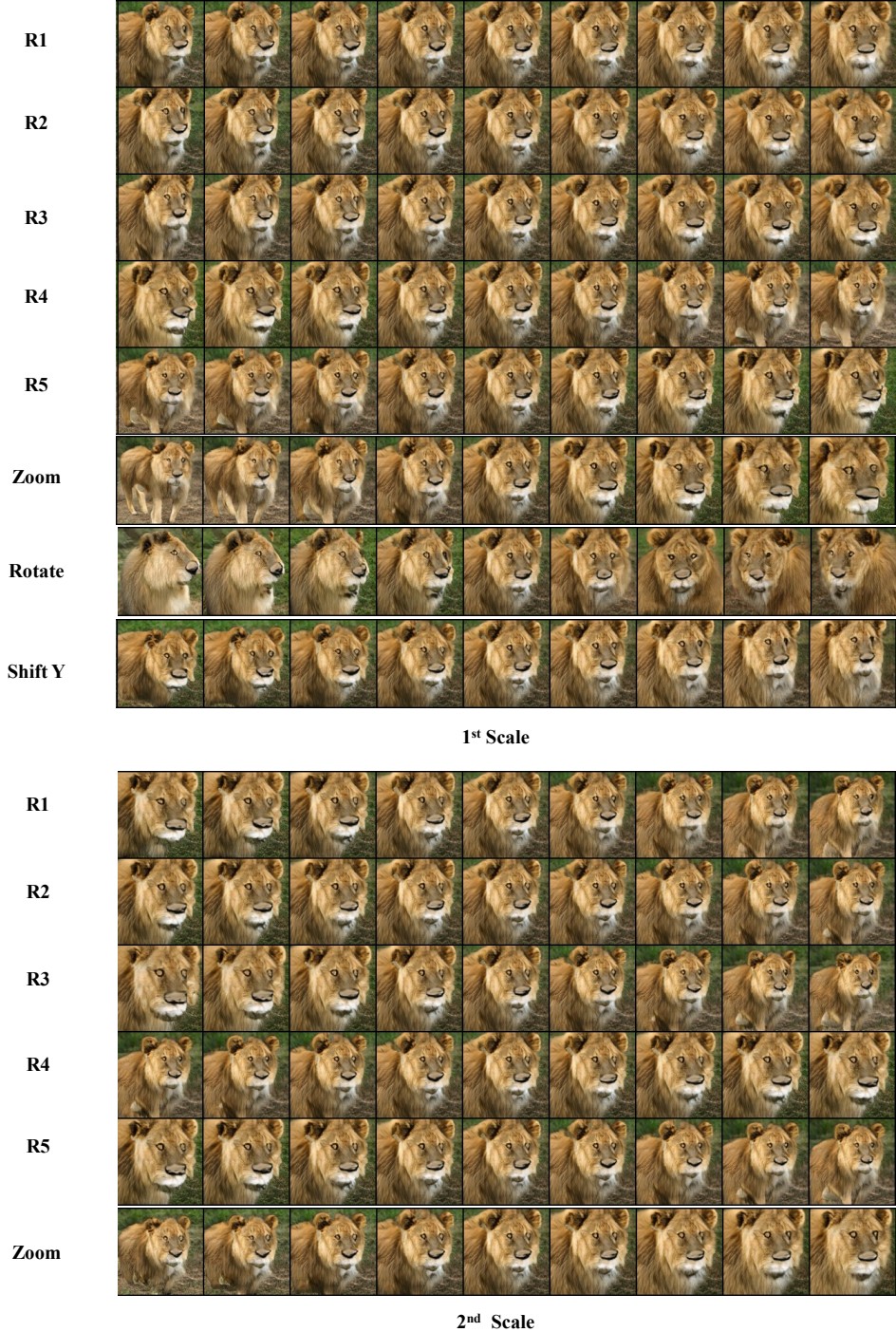

Figure 11: **Our vs. random directions.** We illustrate the effects of five random directions $R1, \ldots, R5$ (normally distributed and scaled to have unit norms) in the first and second scales of BigGAN, in comparison with our principal directions. We can see that each random direction leads to different changes, but it is impossible to associate a single dominant property with each direction. For example, in R5 we can see changes in size, location, and pose. This is while our directions separate those effects into unique paths.

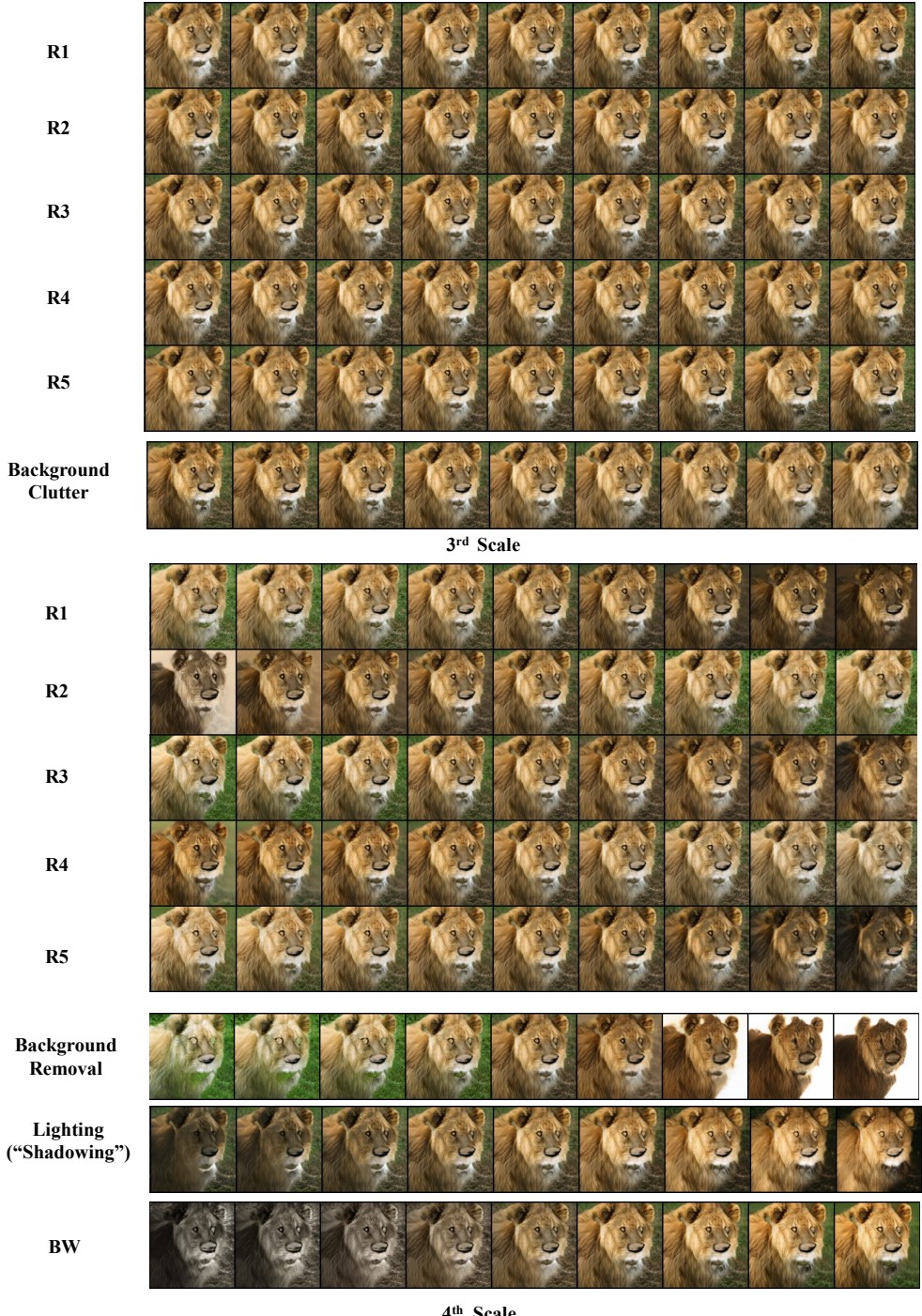

Figure 12: **Our vs. random directions.** We show the effects of five random directions in the third and fourth scales of BigGAN in comparison with our principal directions.

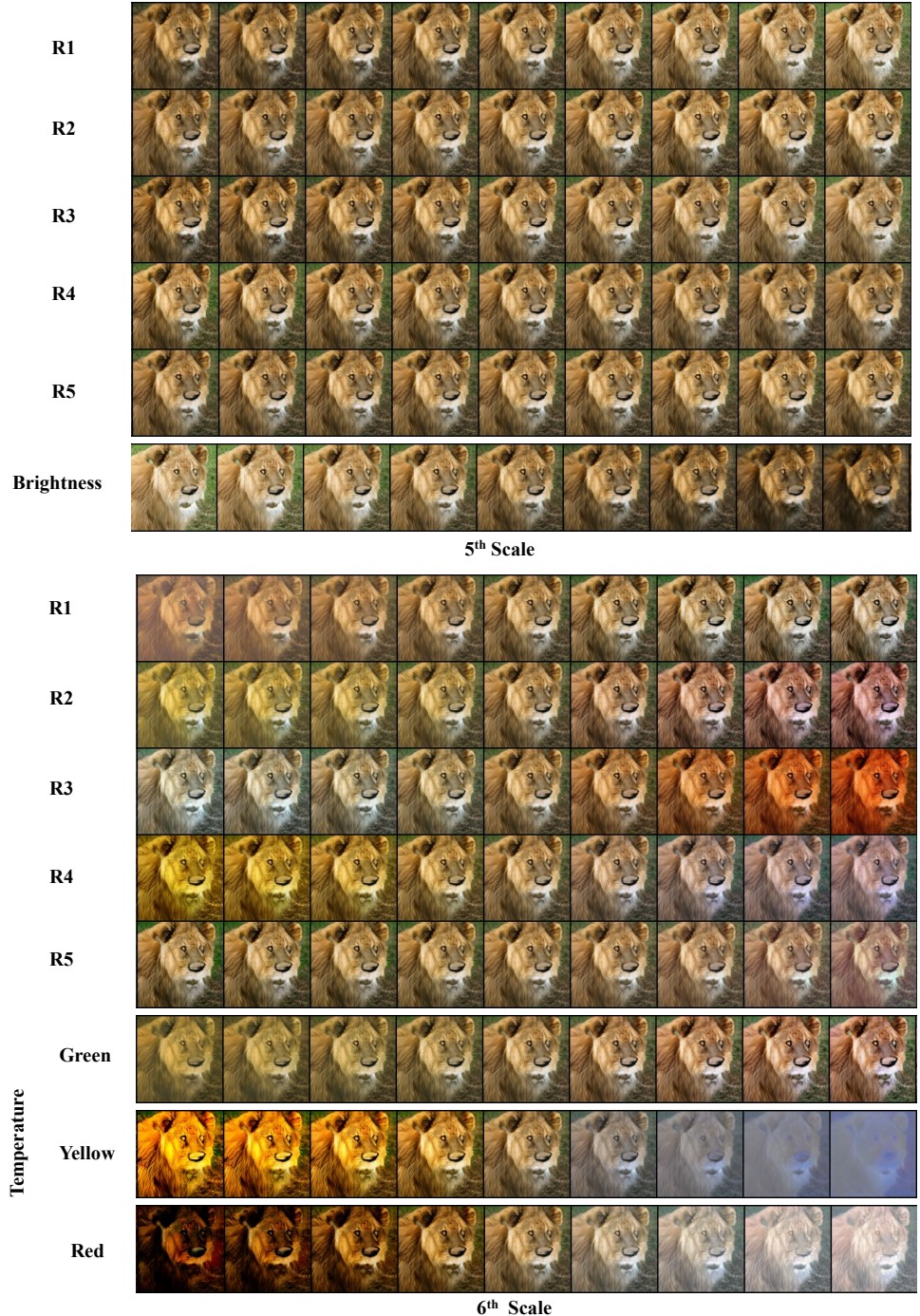

Figure 13: **Our vs. random directions.** We show the effects of five random directions in the fifth and sixth scales of BigGAN in comparison with our principal directions.

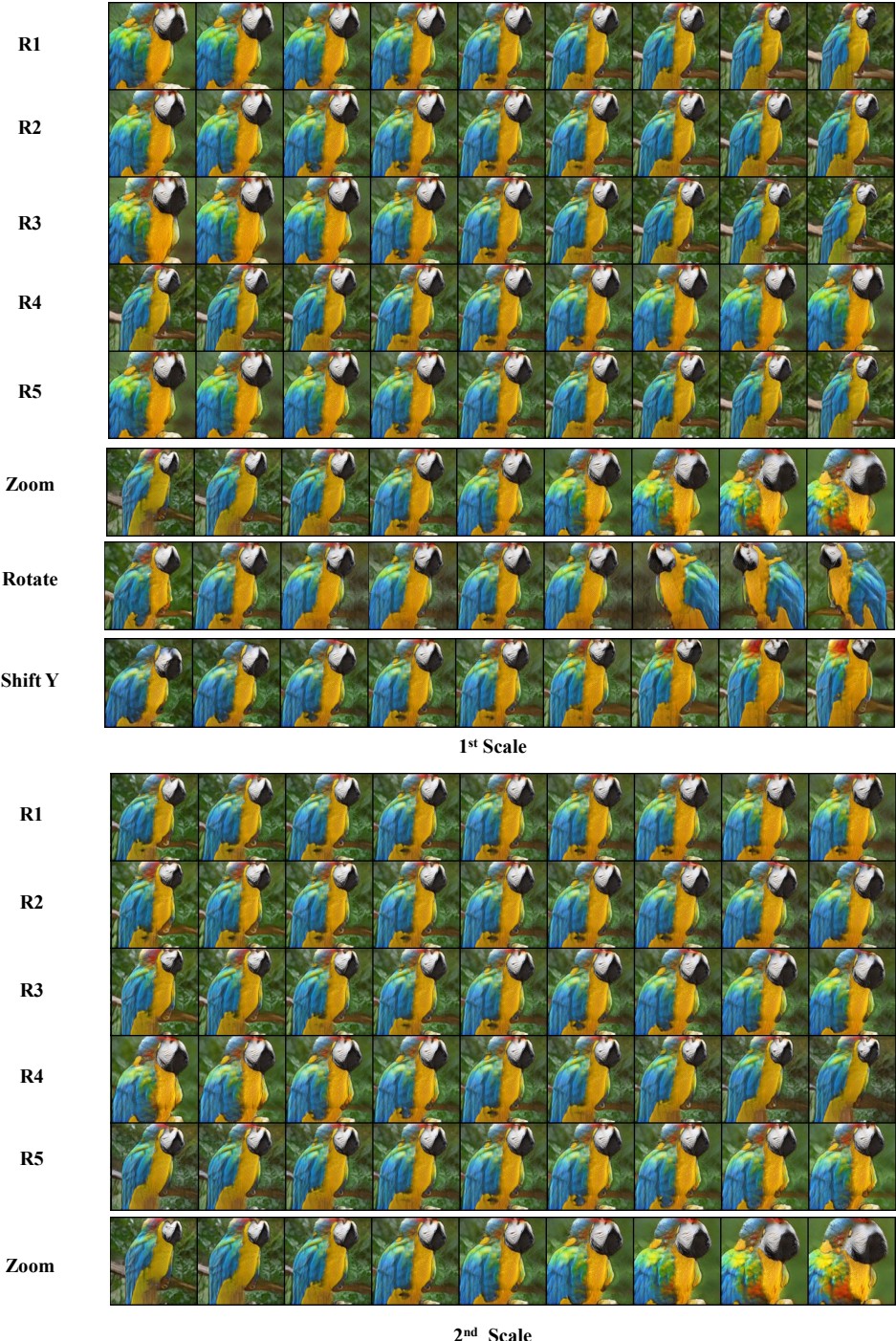

Figure 14: **Our vs. random directions.** We illustrate the effects of five random directions $R1, \ldots, R5$ (normally distributed and scaled to have unit norms) in the first and second scales of BigGAN, in comparison with our principal directions. We can see that each random direction leads to different changes, but it is impossible to associate a single dominant property with each direction. For example, in R5 we can see changes in size, location, and pose. This is while our directions separate those effects into unique paths.

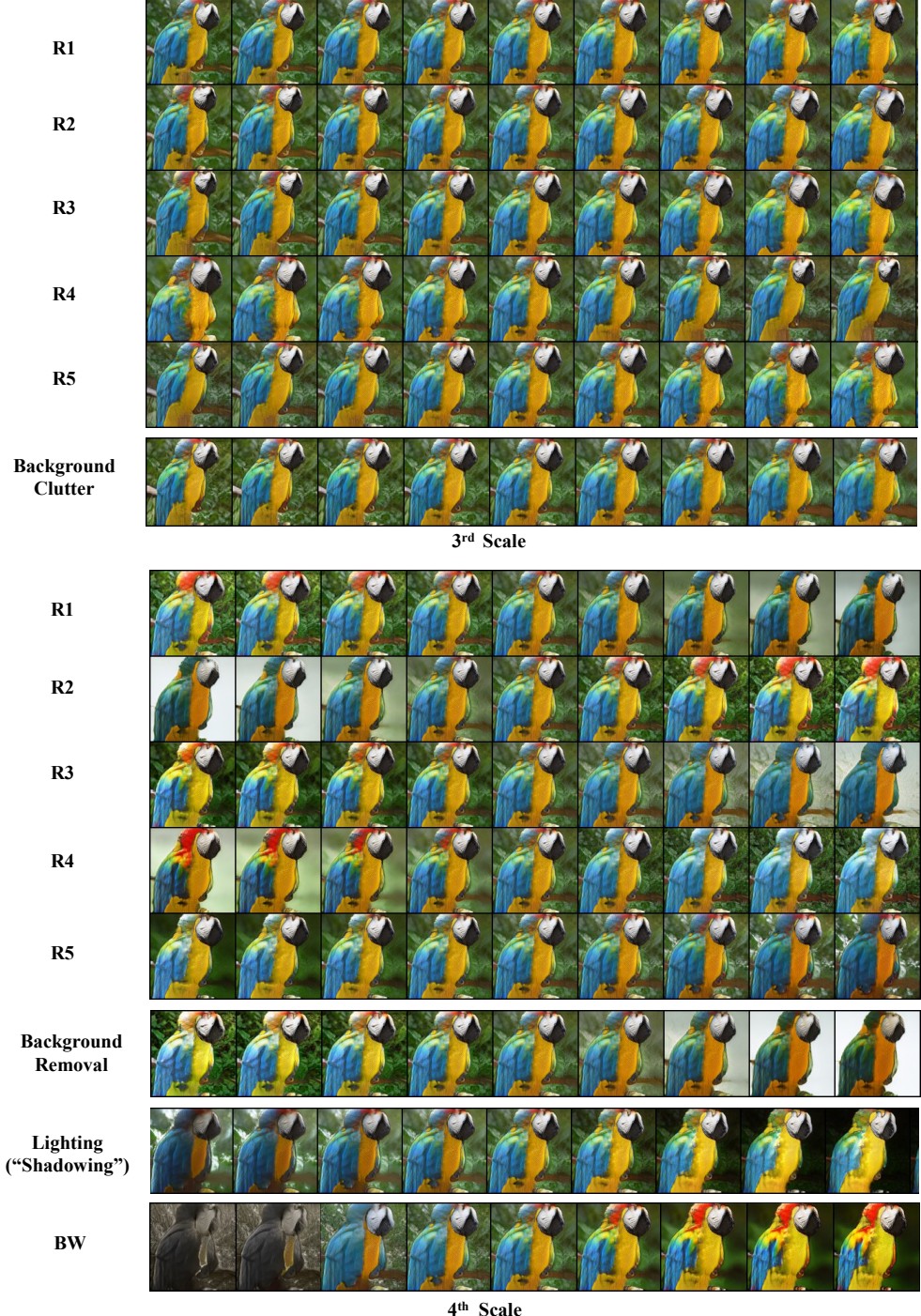

Figure 15: **Our vs. random directions.** We show the effects of five random directions in the third and fourth scales of BigGAN in comparison with our principal directions.

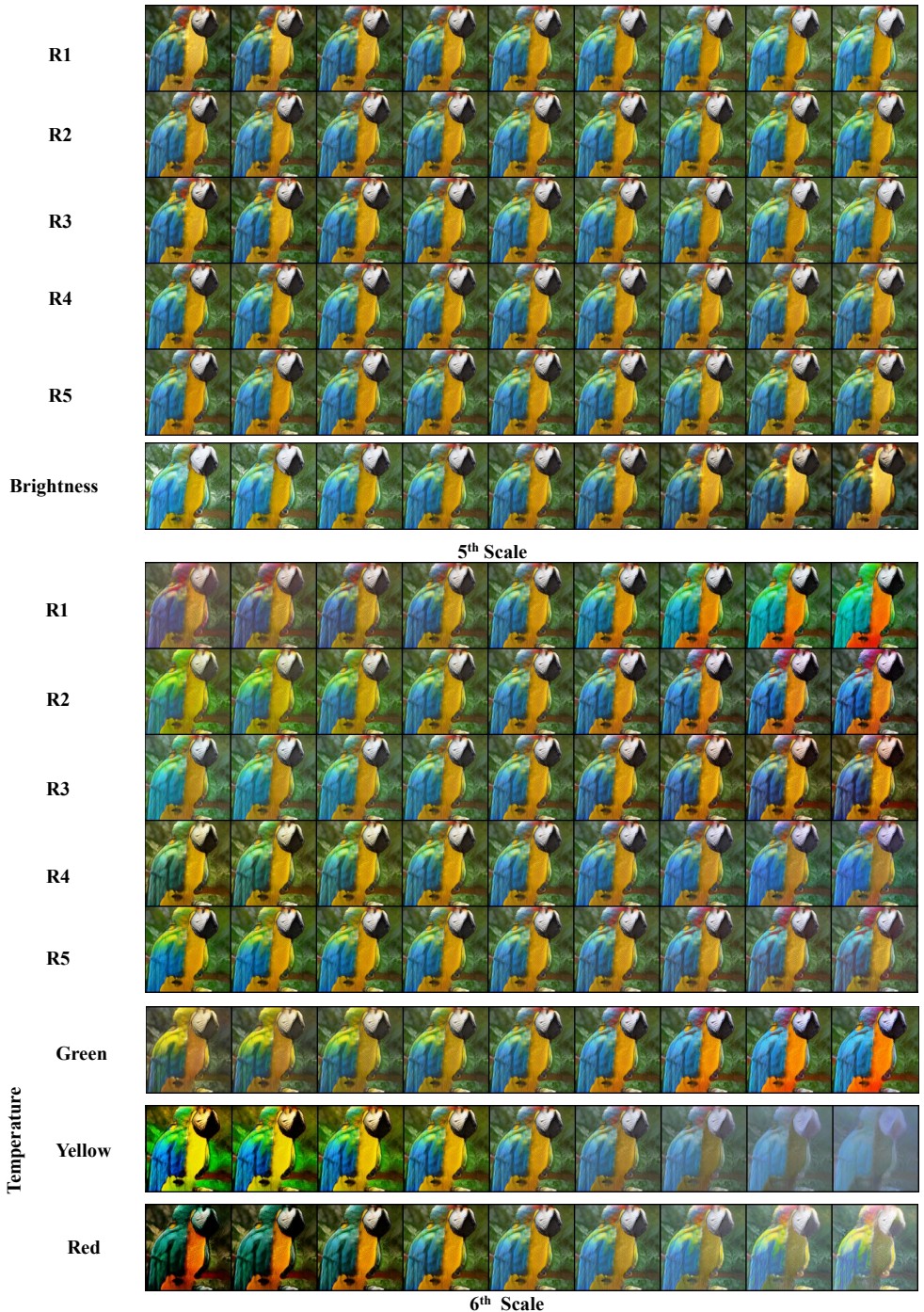

Figure 16: **Our vs. random directions.** We show the effects of five random directions in the fifth and sixth scales of BigGAN in comparison to our principal directions.

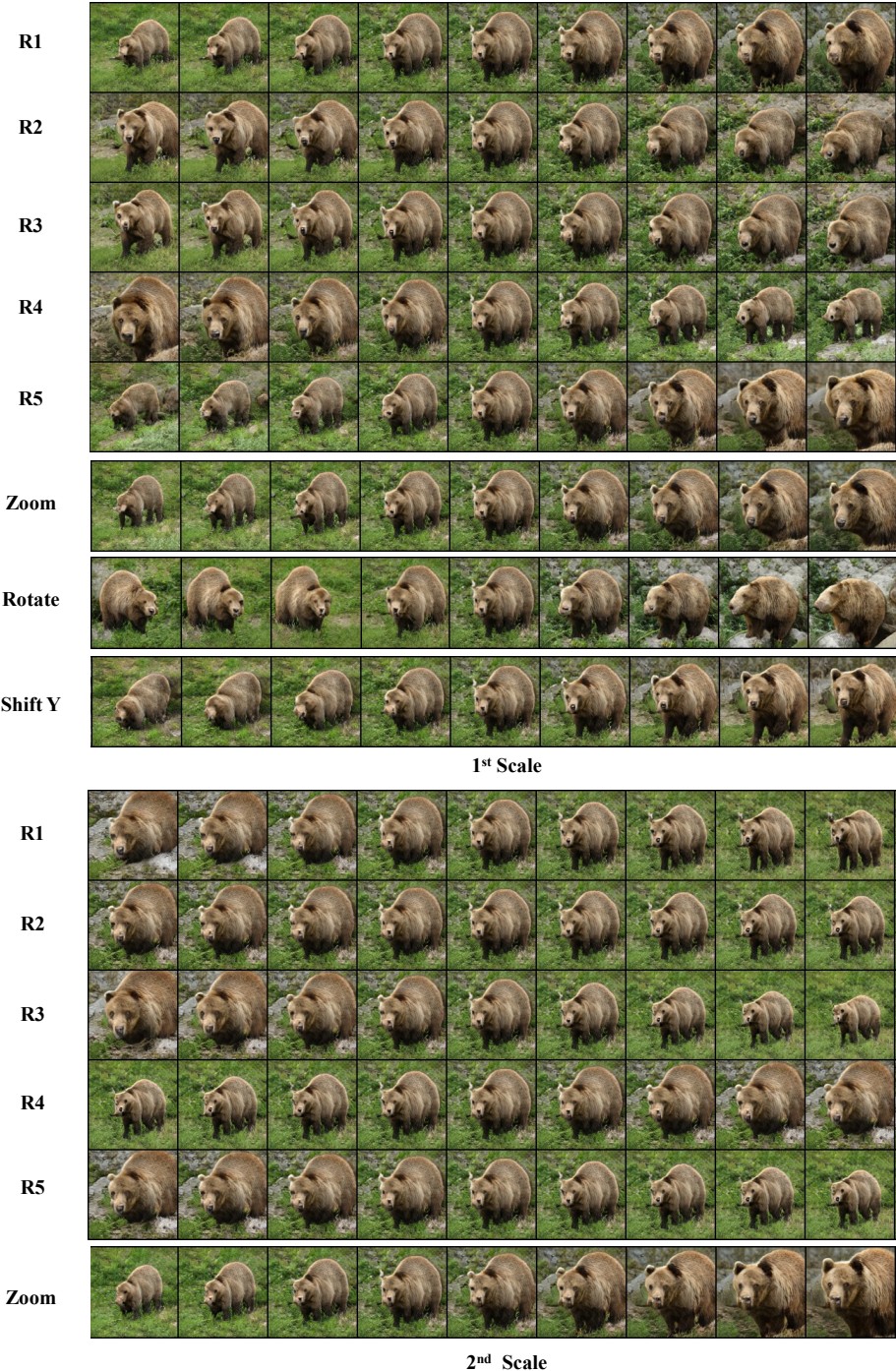

Figure 17: **Our vs. random directions.** We illustrate the effects of five random directions $R1, \ldots, R5$ (normally distributed and scaled to have unit norms) in the first and second scales of BigGAN, in comparison with our principal directions. We can see that each random direction leads to different changes, but it is impossible to associate a single dominant property with each direction. For example, in R5 we can see changes in size, location, and pose. This is while our directions separate those effects into unique paths.

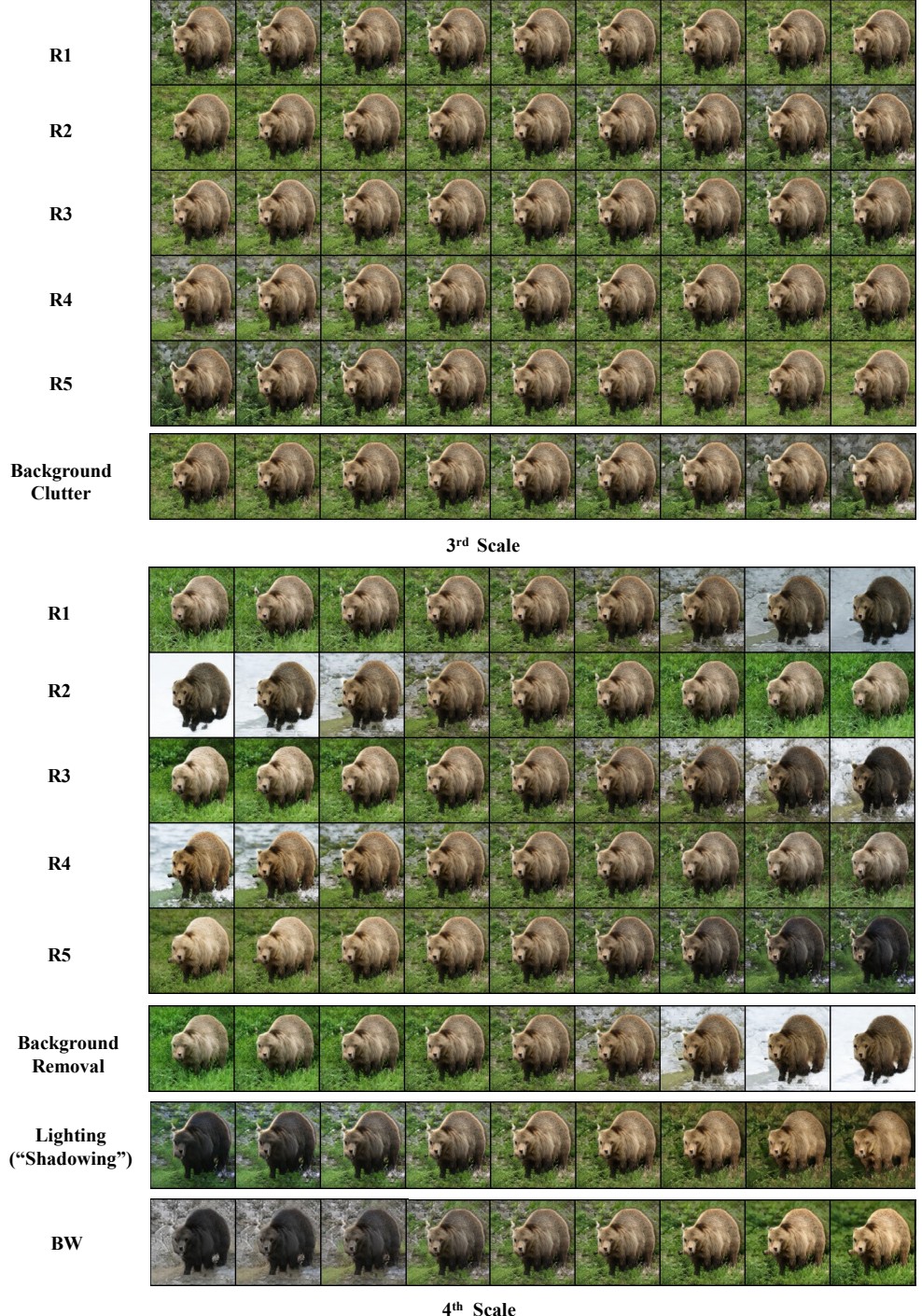

Figure 18: **Our vs. random directions.** We show the effects of five random directions in the third and fourth scales of BigGAN in comparison with our principal directions.

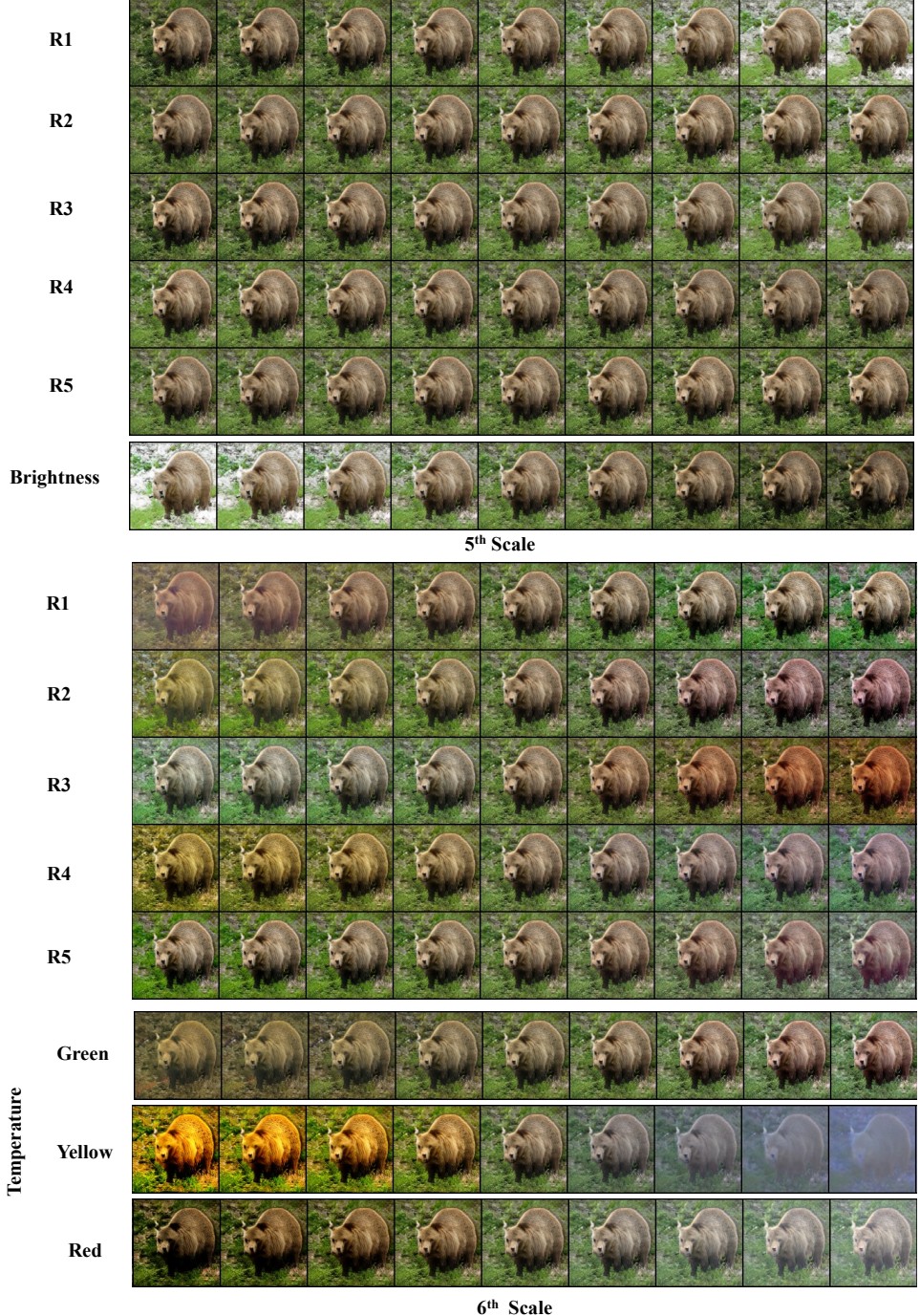

Figure 19: **Our vs. random directions.** We show the effects of five random directions in the fifth and sixth scales of BigGAN in comparison to our principal directions.

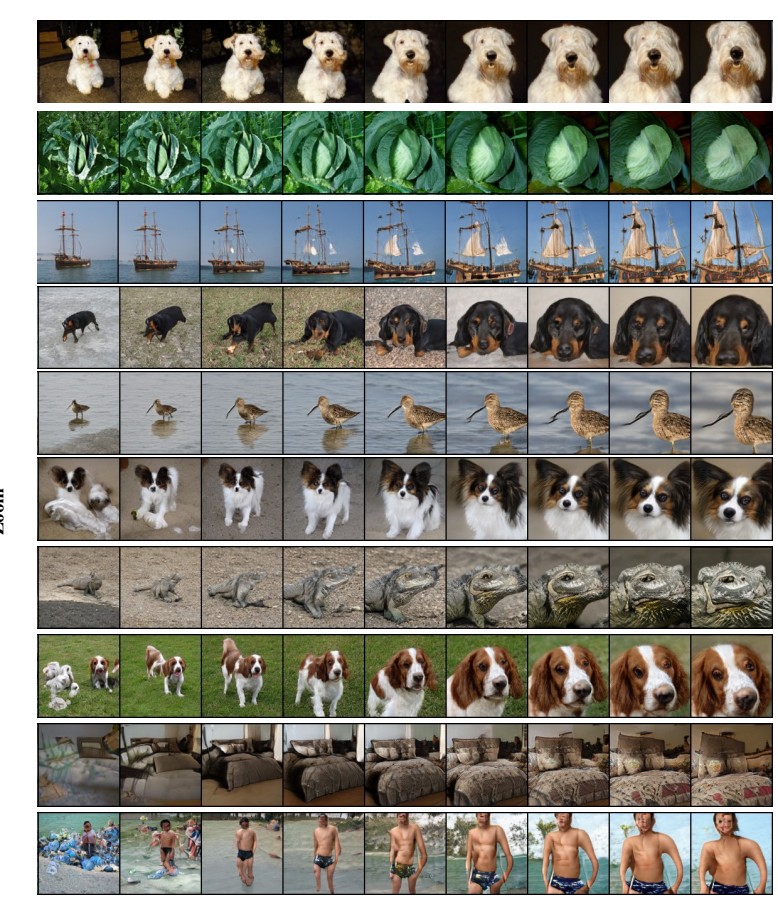

Figure 20: 1st principal direction of the first scale in BigGAN

**Scale 1, Direction 1
Shift up**

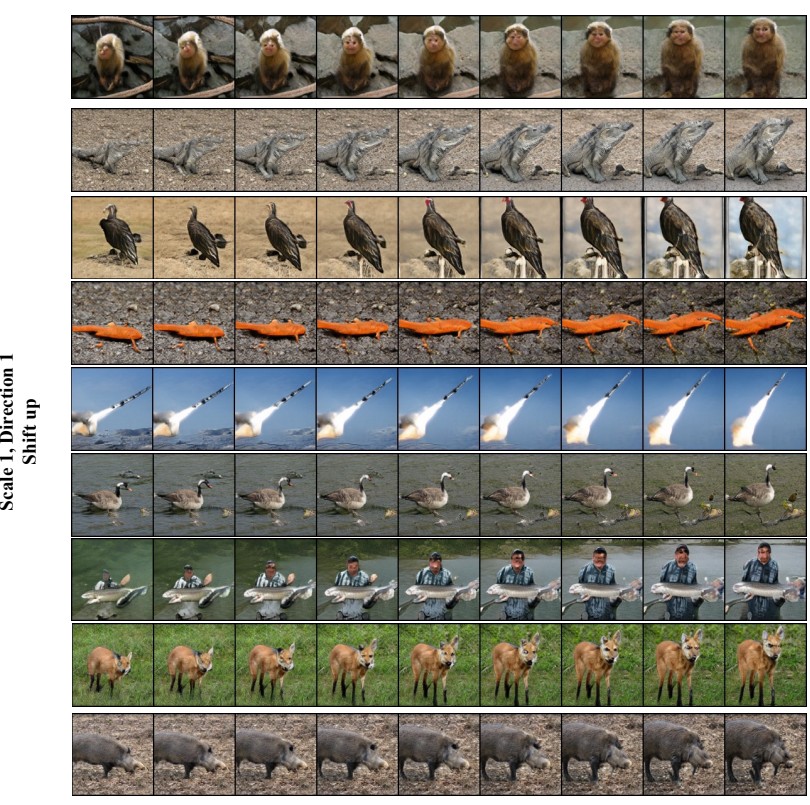

Figure 21: 2nd principal direction of the first scale in BigGAN

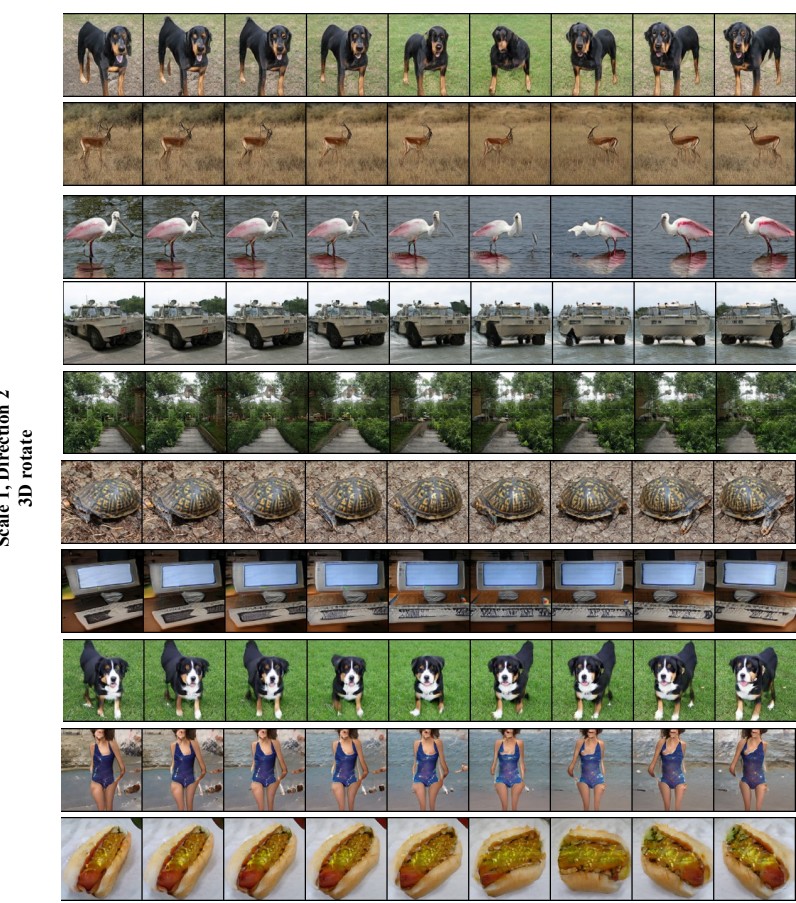

Figure 22: 3rd principal direction of the first scale in BigGAN

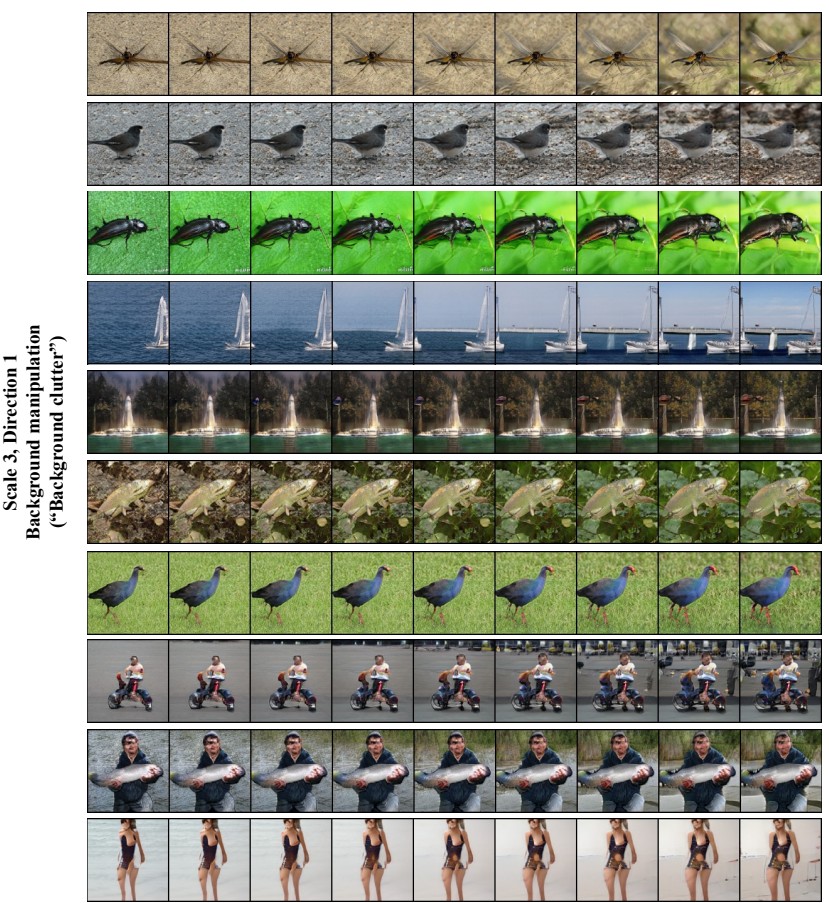

Figure 23: 1st principal direction of the third scale in BigGAN

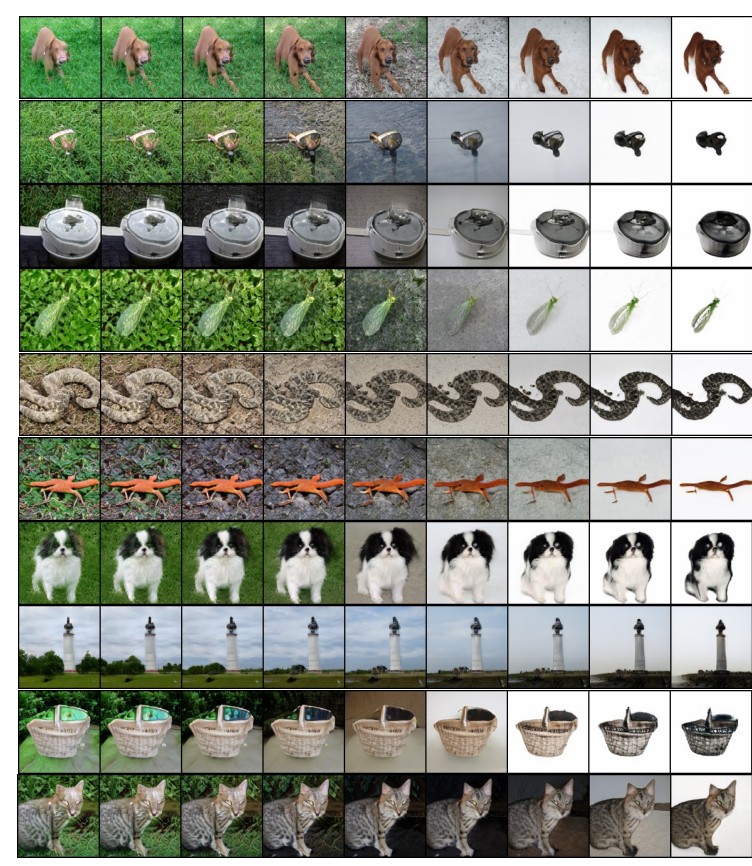

Figure 24: First principal direction of the fourth scale in BigGAN

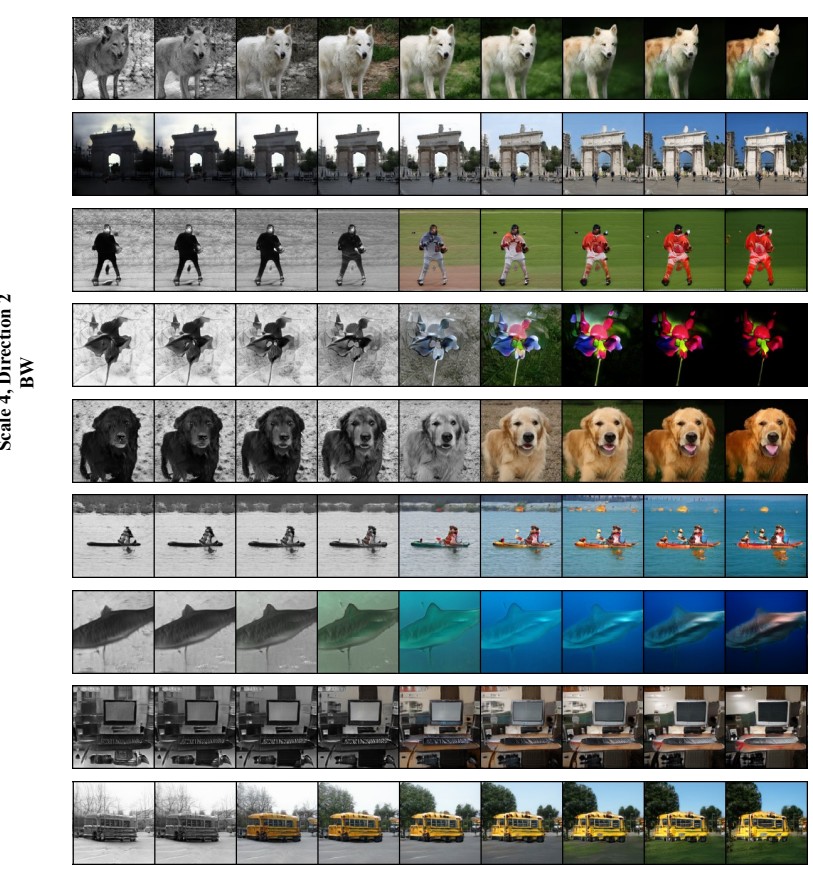

Figure 25: Second principal direction of the fourth scale in BigGAN

**Scale 4, Direction 3
Lighting ("Shadowing")**

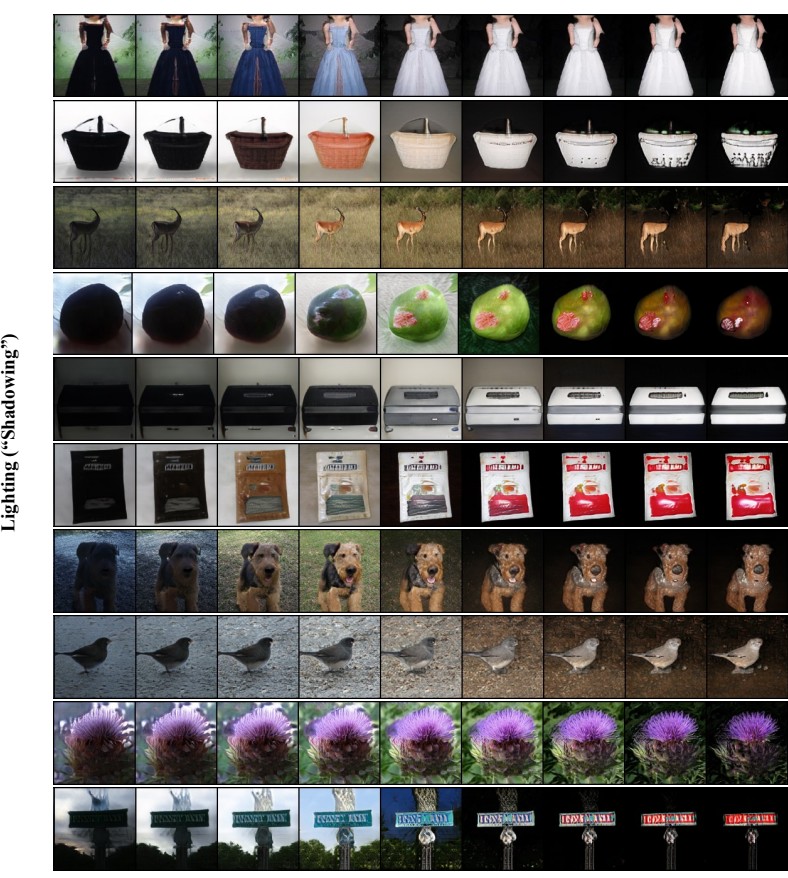

Figure 26: Third principal direction of the fourth scale in BigGAN

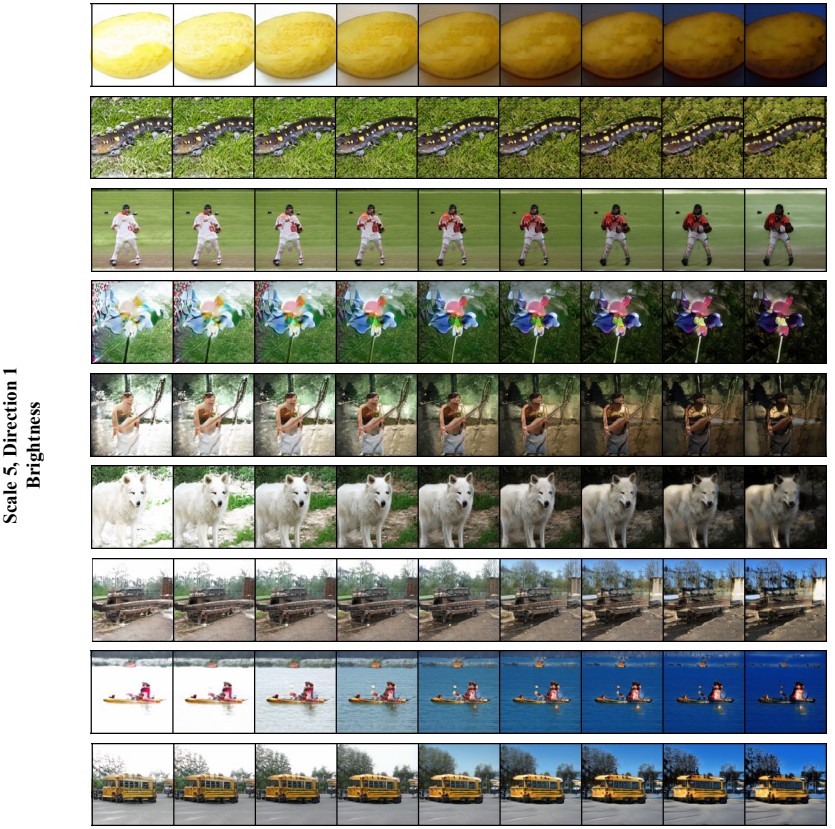

Figure 27: First principal direction of the fifth scale in BigGAN

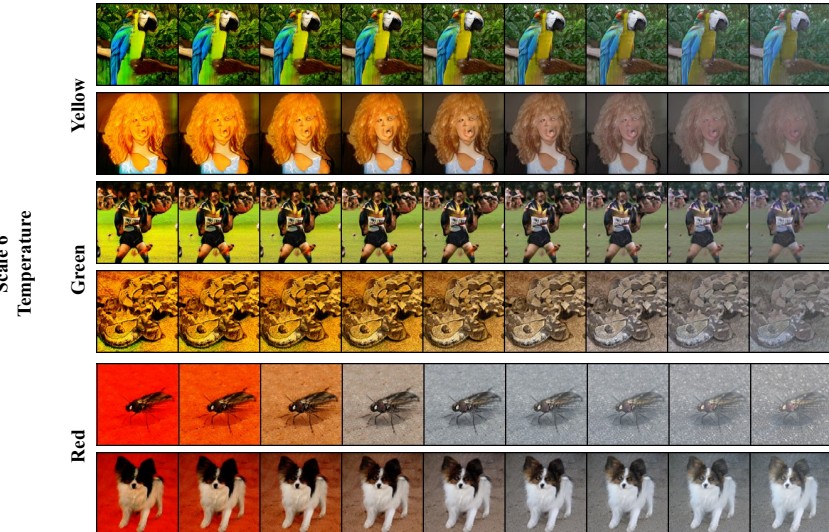

Figure 28: First three principal direction of the sixth scale in BigGAN

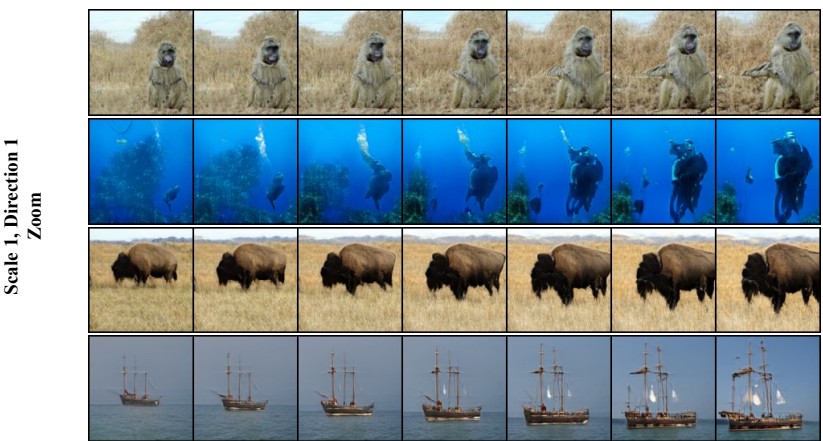

Figure 29: First principal direction of scale 1 in BigGAN (small circle walks).

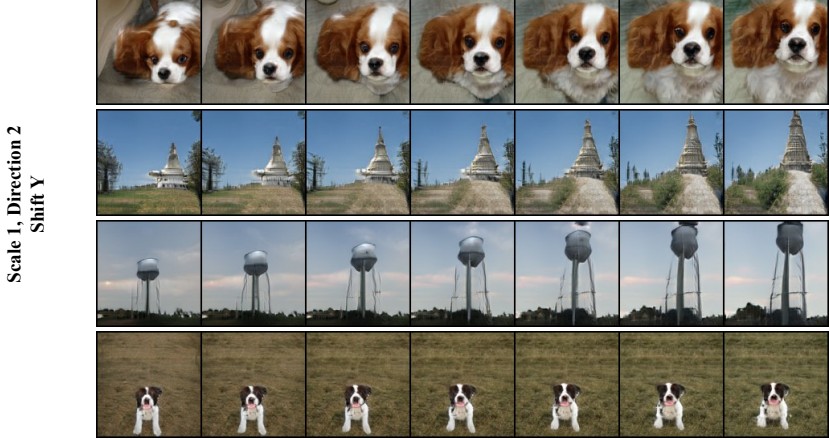

Figure 30: Second principal direction of scale 1 in BigGAN (small circle walks).

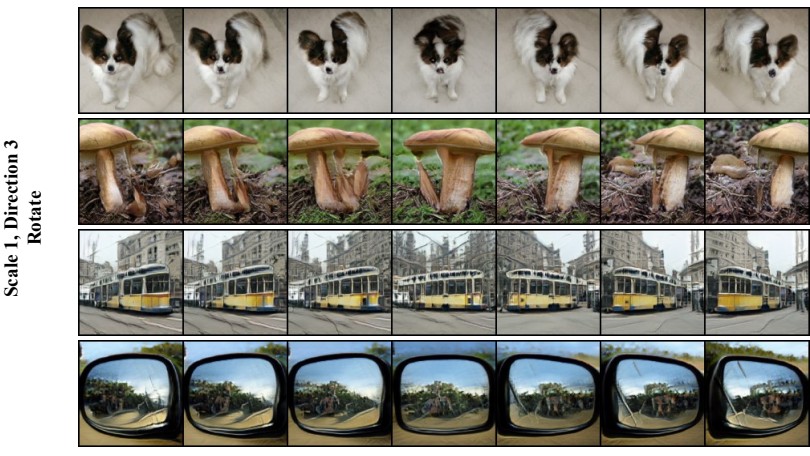

Figure 31: Third principal direction of scale 1 in BigGAN (small circle walks).

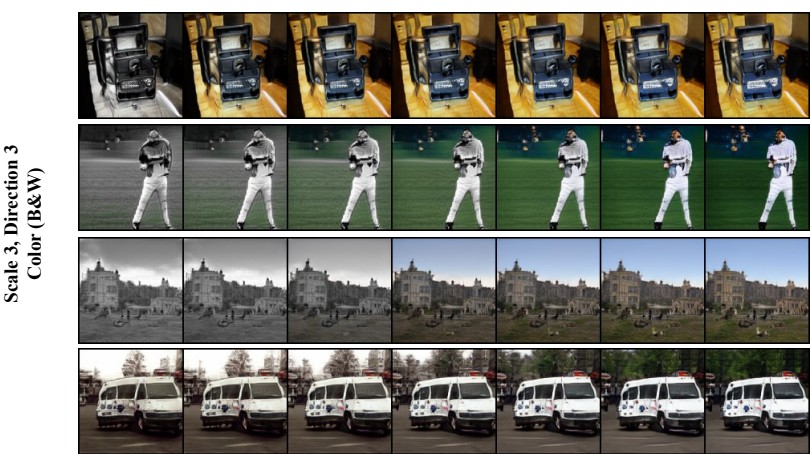

Figure 32: Third principal direction of scale 4 in BigGAN (small circle walks).

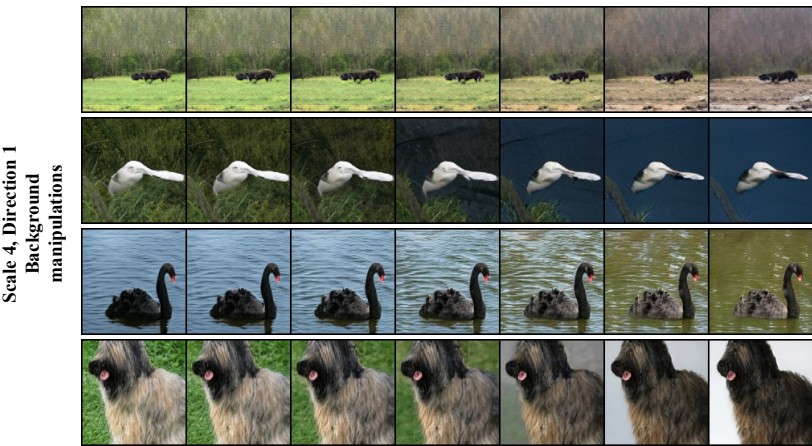

Figure 33: First principal direction of scale 4 (small circle walks). When walking enough steps in the linear direction, a total background removal is observed (see Fig.15. However, it might come with a slight change of object colors. Therefore, we will not constantly see it within the small circle framework (see last image in that bulk in comparison to the other 3).

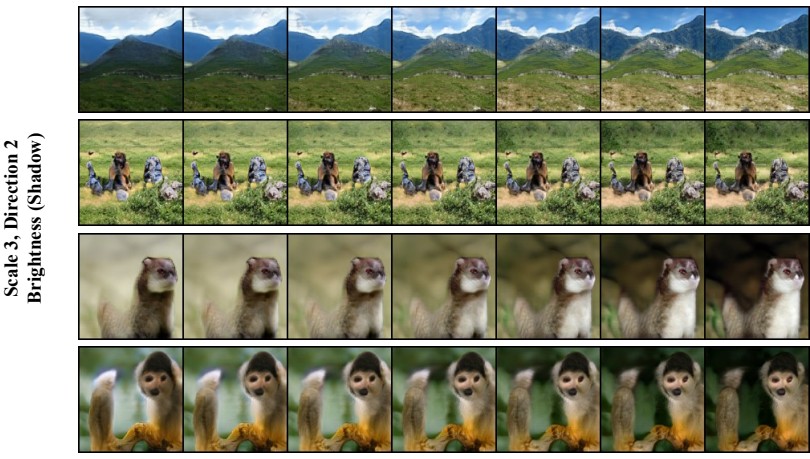

Figure 34: Second principal direction of scale 4 in BigGAN (small circle walks).

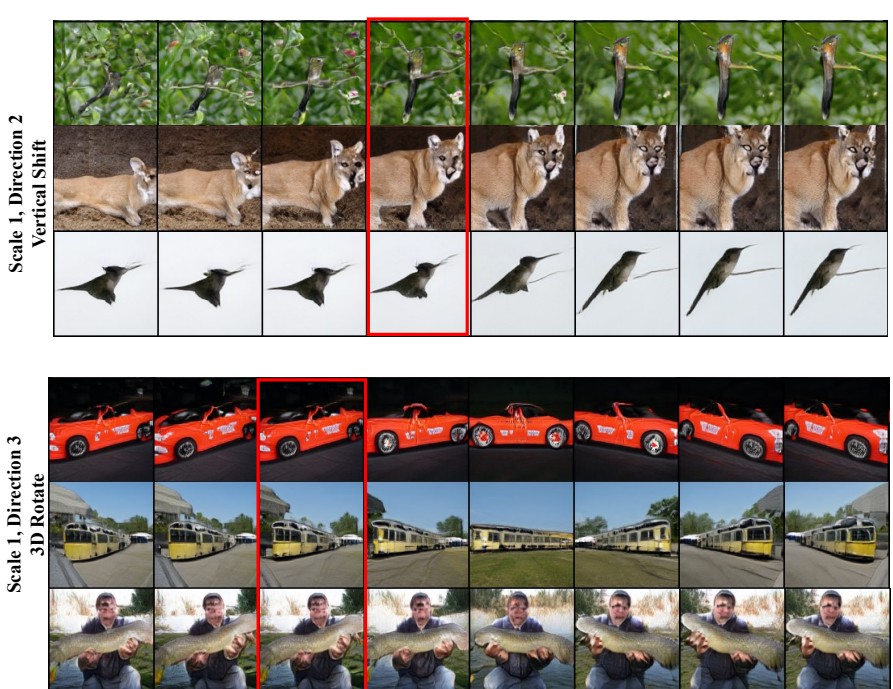

Figure 35: Chosen principal direction of scale 3 in BigGAN (small circle walks). When the initial generated image is not at the middle of the path, we need to take different number of steps to each side.

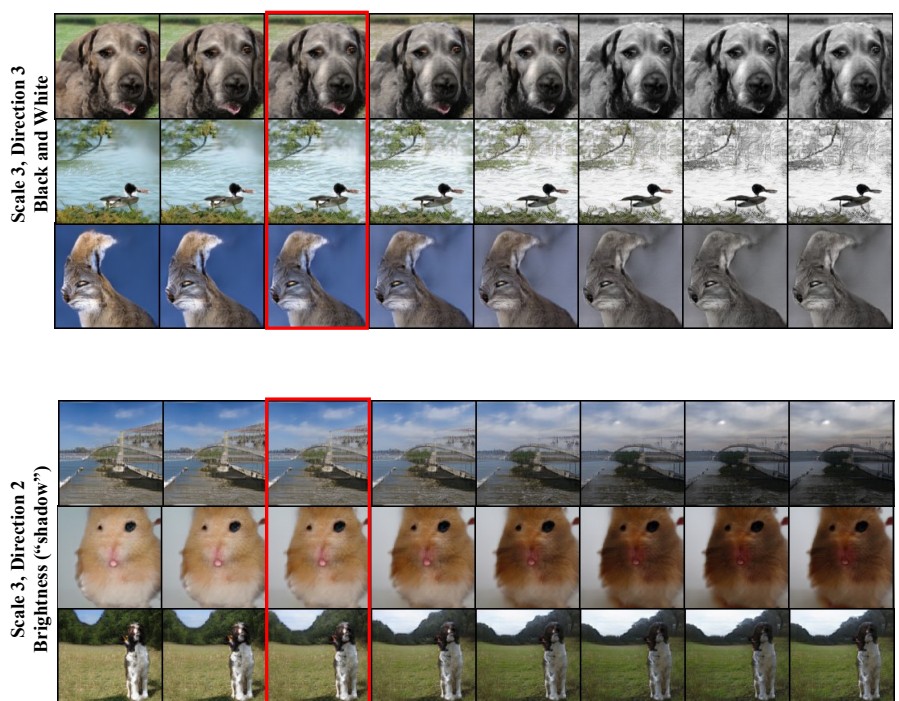

Figure 36: Chosen principal direction of scale 1 in BigGAN (small circle walks).When the initial generated image is not at the middle of the path, we need to take different number of steps at each side.

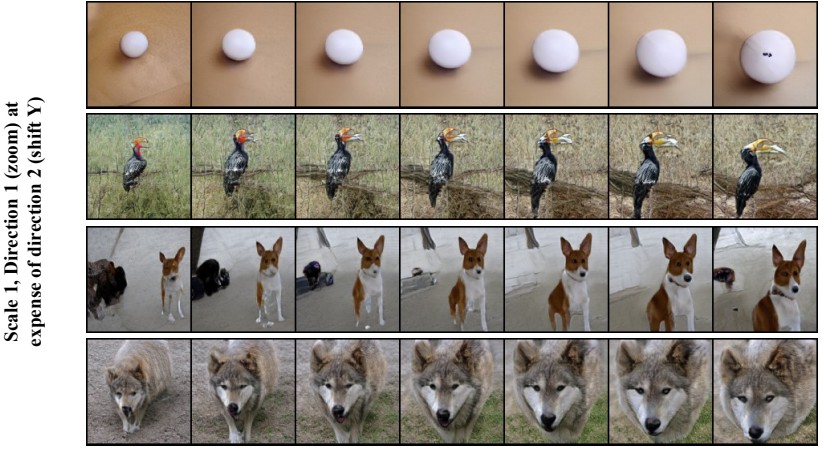

Figure 37: Modifying the second principal direction of scale 1 on the expense of the first principal direction of that scale in BigGAN (small circle walks).

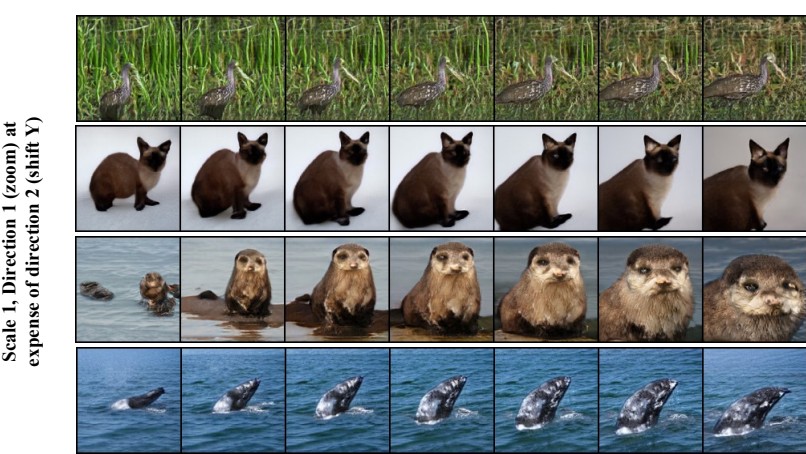

Figure 38: Modifying the second principal direction of scale 1 on the expense of the first principal direction of that scale in BigGAN (small circle walks).

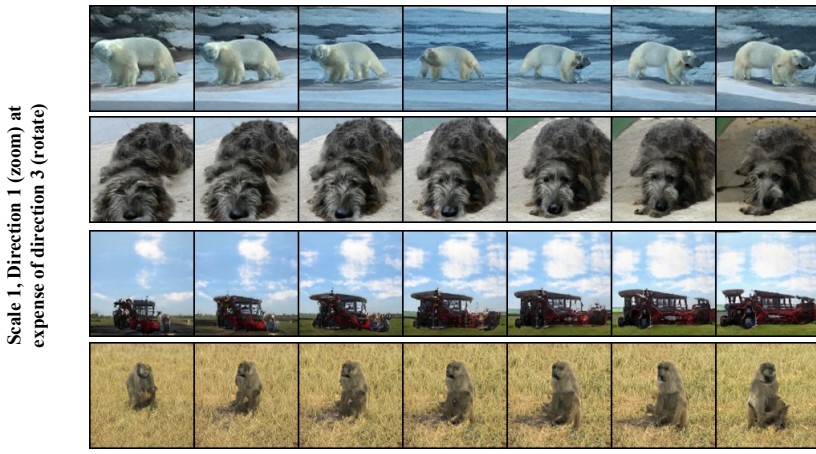

Figure 39: Modifying the third principal direction of scale 1 on the expense of the first principal direction of that scale in BigGAN (small circle walks).

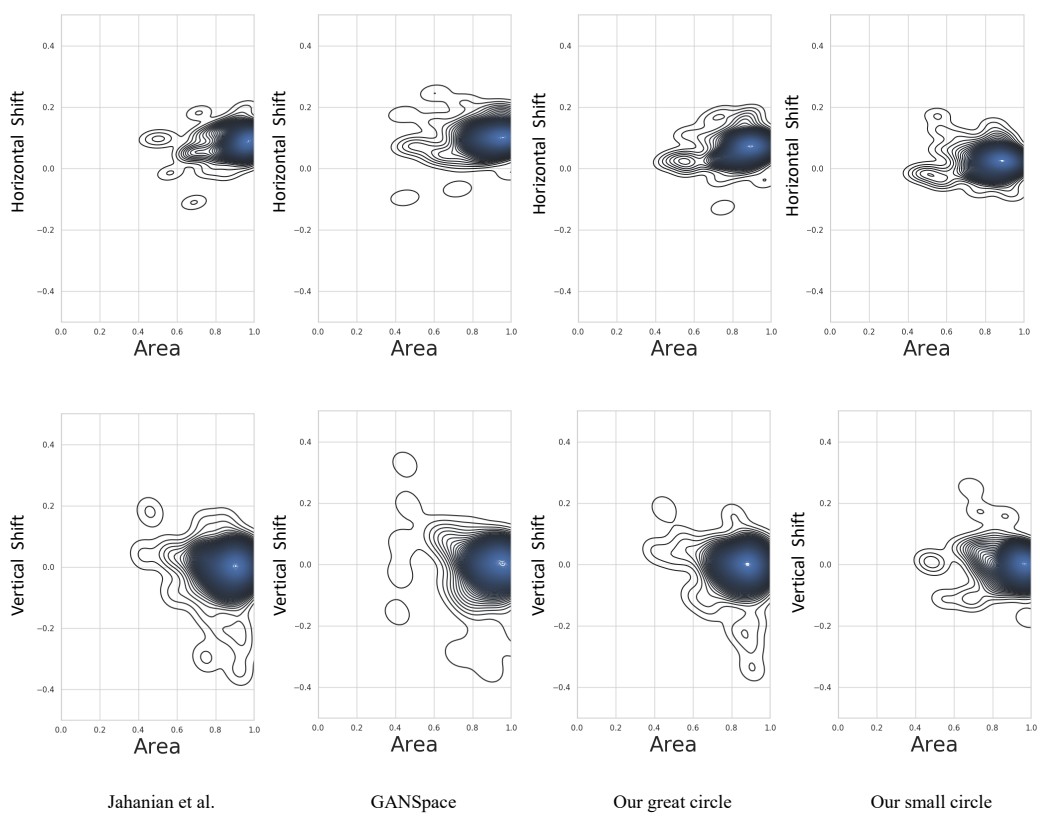

Jahanian et al.        GANSpace        Our great circle        Our small circle

Figure 40: **Second order dataset biases.** We explore the coupling between zoom and horizontal translation (top) and zoom and vertical translation (bottom) for Persian cat class in BigGAN-deep. It can be clearly observed that the small circle path exhibits the smallest undesired shifts when increasing the area.

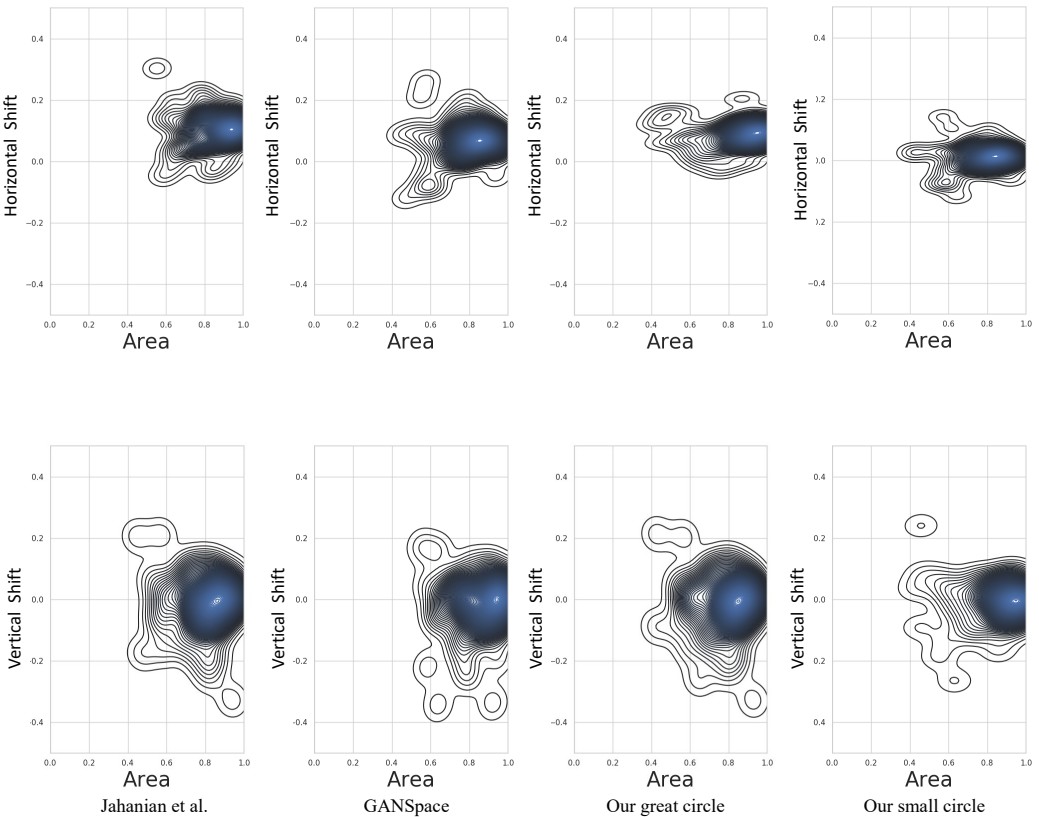

Figure 41: **Second order dataset biases.** We explore the coupling between zoom and horizontal translation (top) and zoom and vertical translation (bottom) for husky dogs class in BigGAN-deep. It can be clearly observed that the small circle path exhibits the smallest undesired shifts when increasing the area.

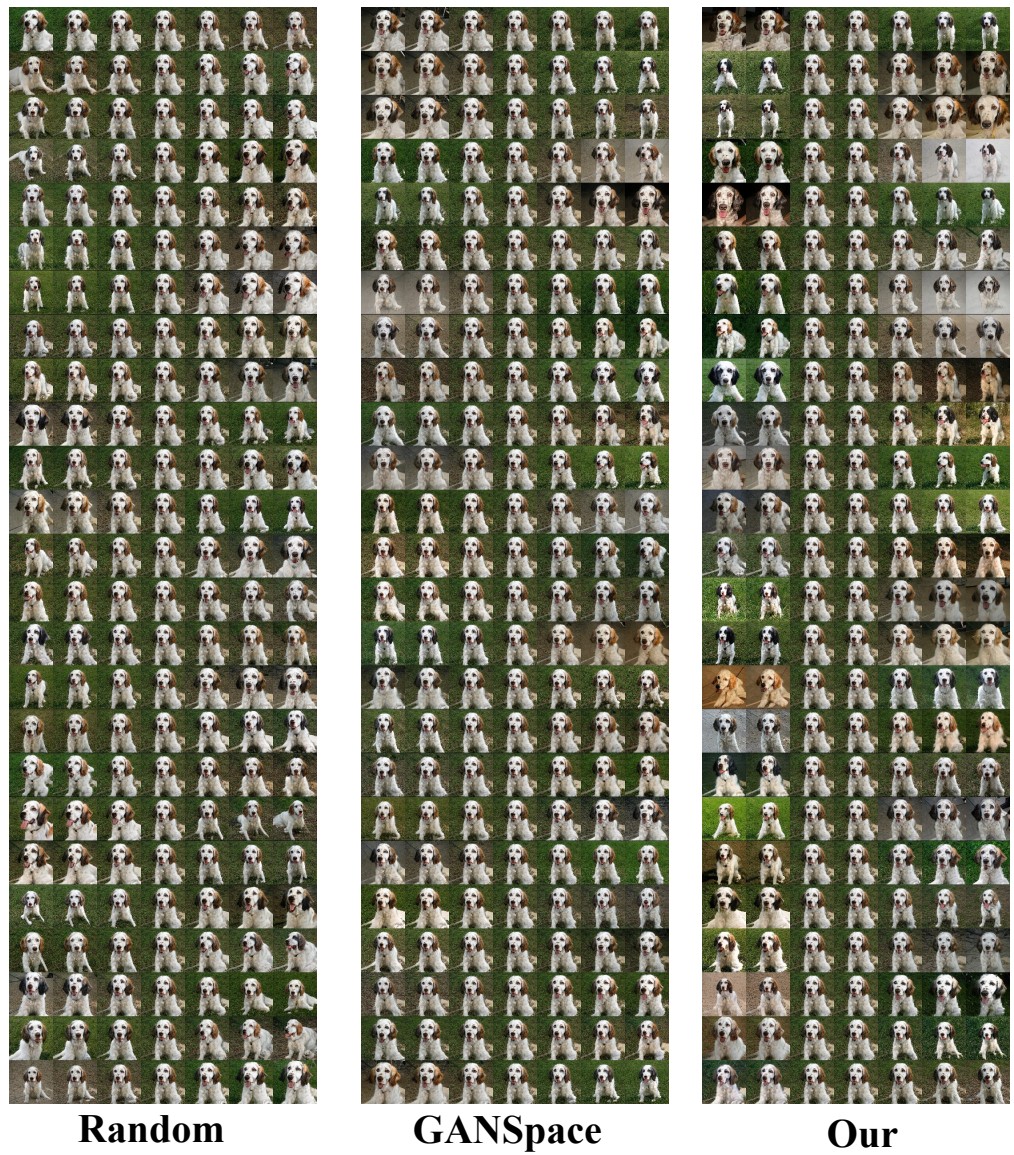

**Random** **GANSpace** **Our**

Figure 42: **Comparison with GANSpace and random directions in BigGAN-deep** (principal vectors 0-25). The image at the center of each block is the original image. We linearly added the vectors with equal steps. Both directions are normalized to have unit-norms. We can see that our trajectories induce a stronger change than those of Härkönen et al. (2020). The averaged LPIPS variance is 0.036 and 0.059 for Härkönen et al. (2020) and our method, respectively.

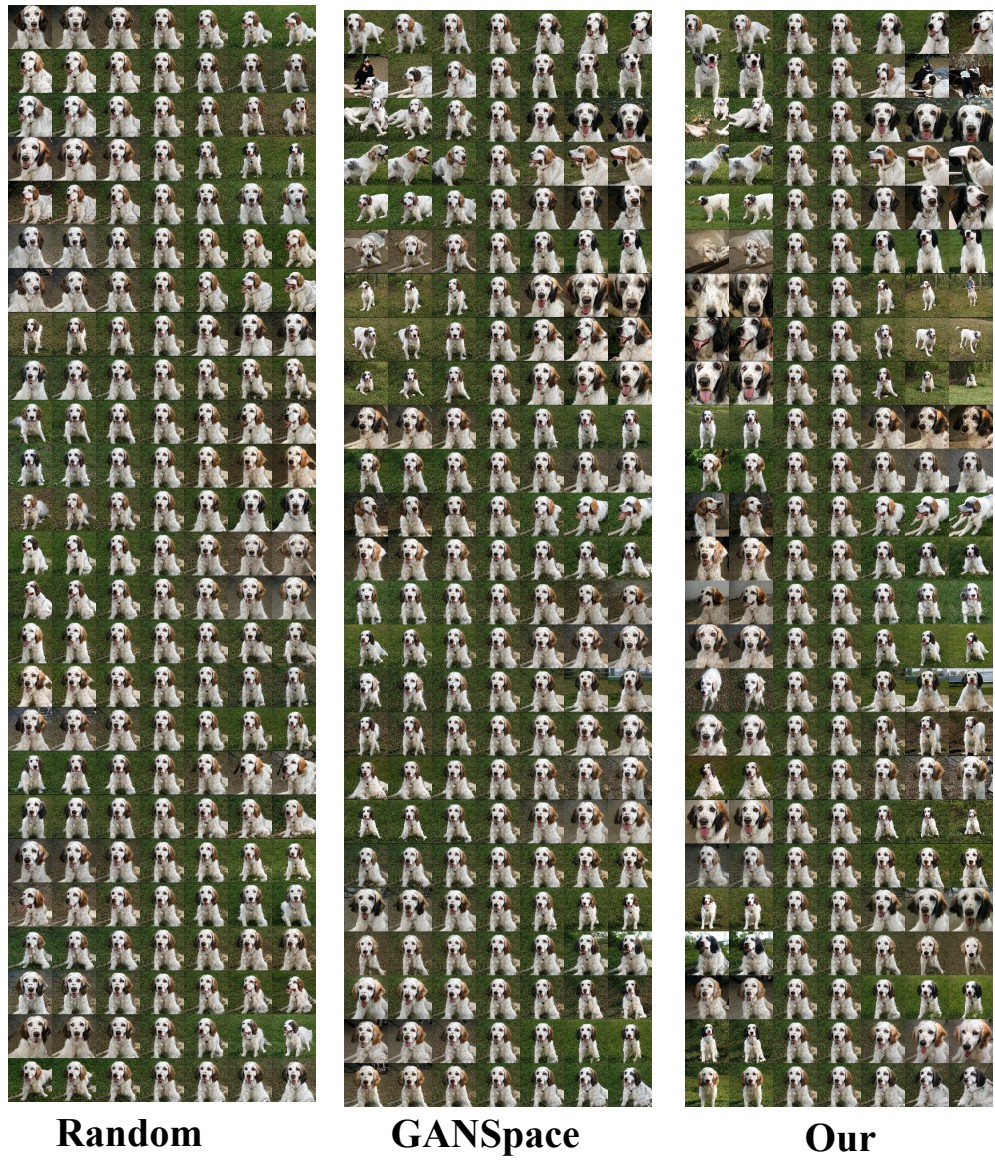

**Random**          **GANSpace**          **Our**

Figure 43: **Comparison with GANSpace and random directions in BigGAN-deep** (principal vectors 25-50). The image at the center of each block is the original image. We linearly added the vectors with equal steps. Both directions are normalized to have unit-norms. It can be observed that our trajectories induce stronger change than those of Härkönen et al. (2020). The averaged LPIPS variance is 0.03 and 0.049 for Härkönen et al. (2020) and our method, respectively.

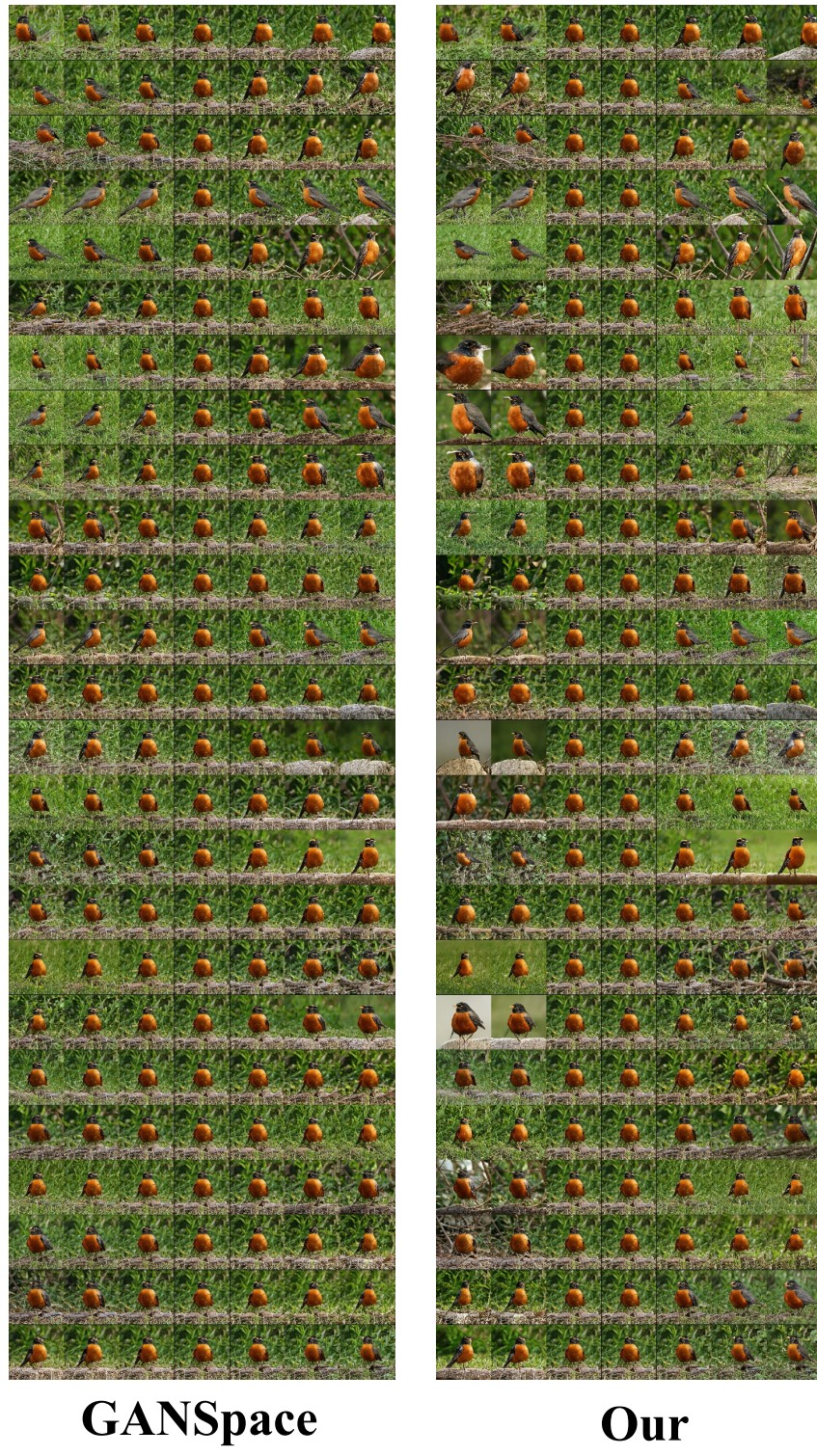

**GANSpace**      **Our**

Figure 44: **Comparison with GANSpace in BigGAN-deep** (principal directions 0-25). The image at the center of each block is the original image. We linearly added the vectors with equal steps. Both directions are normalized to have unit-norms. It can be observed that our trajectories induce stronger change than those of Härkönen et al. (2020).

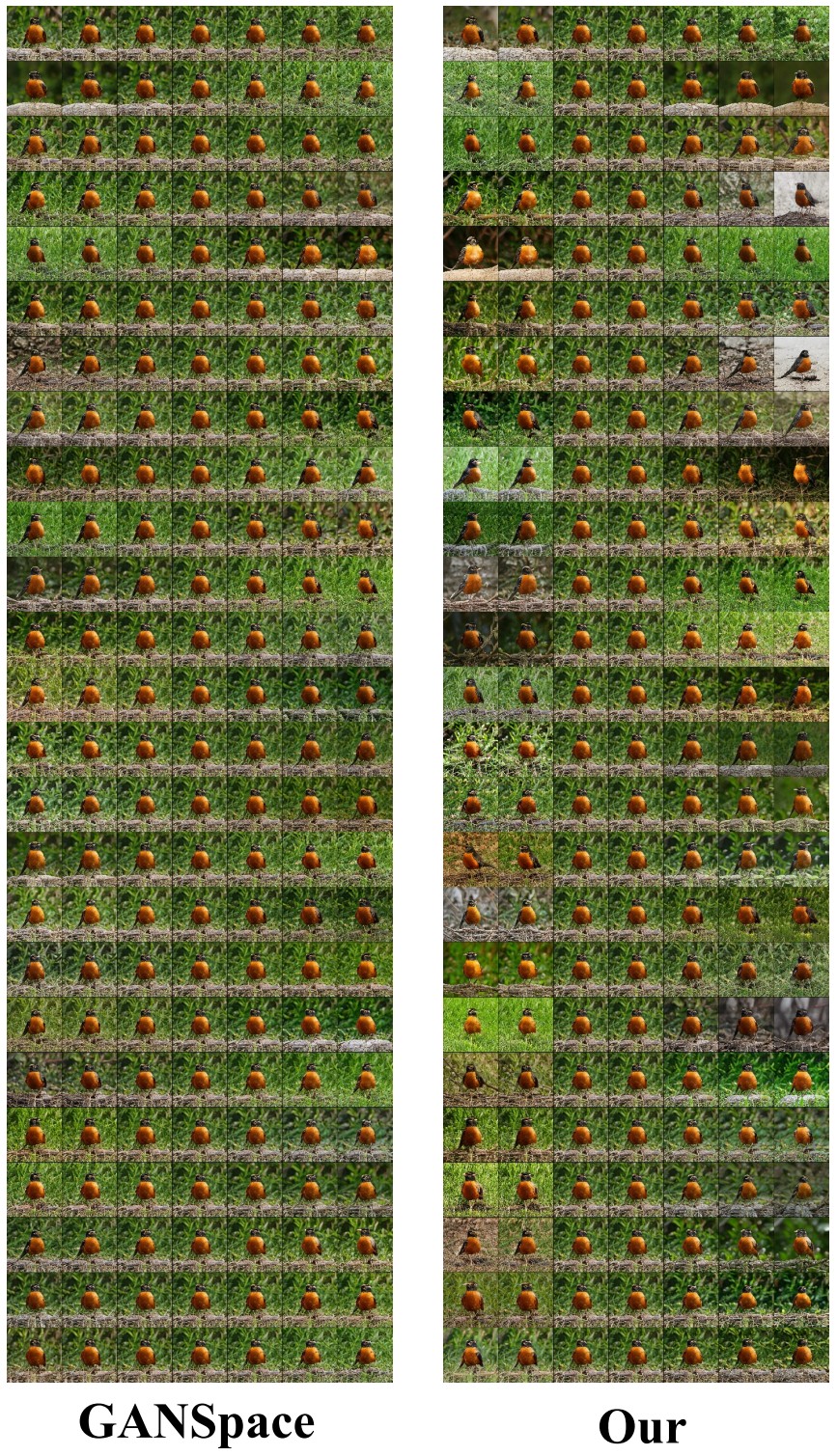

**GANSpace**    **Our**

Figure 45: **Comparison with GANSpace in BigGAN deep** (principal directions 25-50).

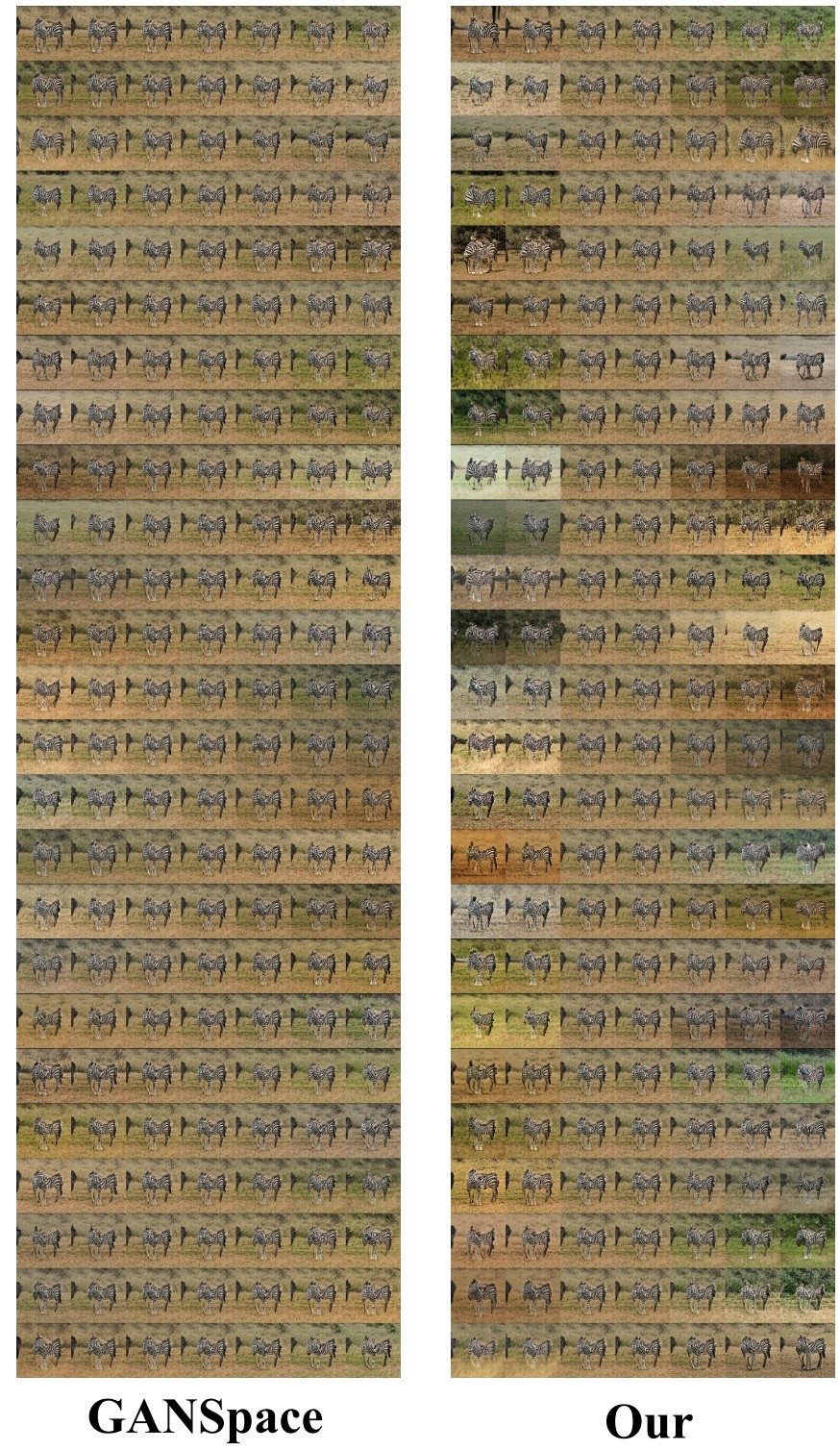

GANSpace                    Our

Figure 46: **Comparison with GANSpace in bigGAN deep** - (principal directions 25-50).

## - "Show horizon" +

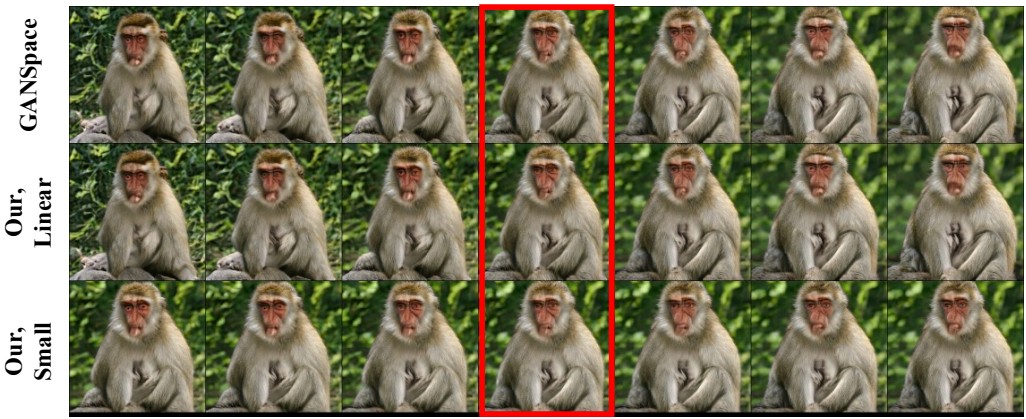

Figure 47: **Comparison with Härkönen et al. (2020)**. An example for the "show horizon" direction which we apply to edit only layers 1-5 (Härkönen et al., 2020) in BigGAN-deep 512. We can see that our linear directions achieve similar effects to those of GANSpace (blurring the background). However, in both cases, we can also see slight changes in the object size and pose. On the other hand, when using our small circle walk, we keep the same size and pose.

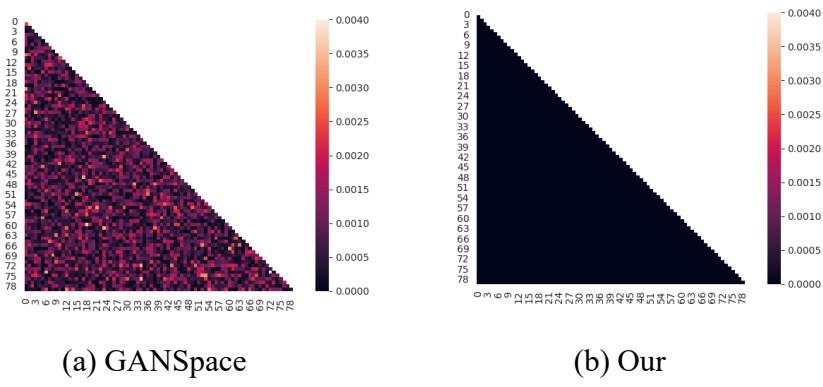

(a) GANSpace

(b) Our

Figure 48: **Evaluating orthogonality**. We show absolute value of the correlation between every two directions among the first 80 directions in GANspace and in our method for BigGAN-deep-512.

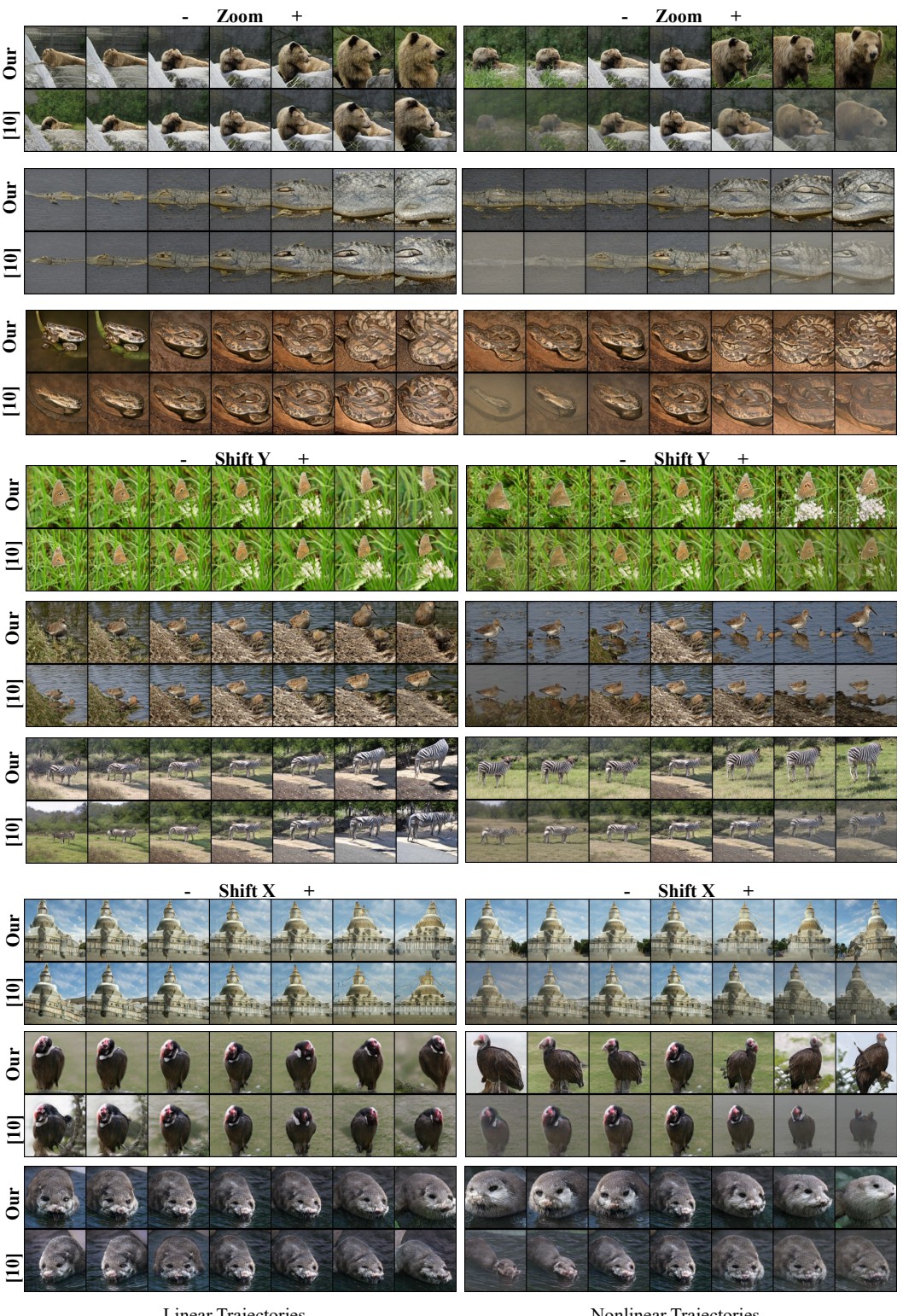

Figure 49: User prescribed transformations with BigGAN.

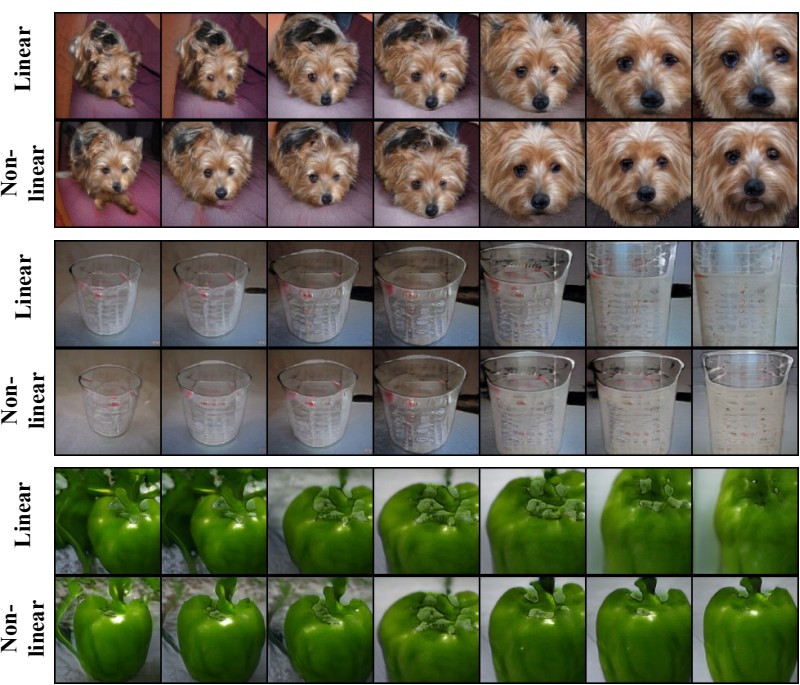

Figure 50: User prescribed zoom with BigGAN. Our method, linear vs. non-linear trajectories.

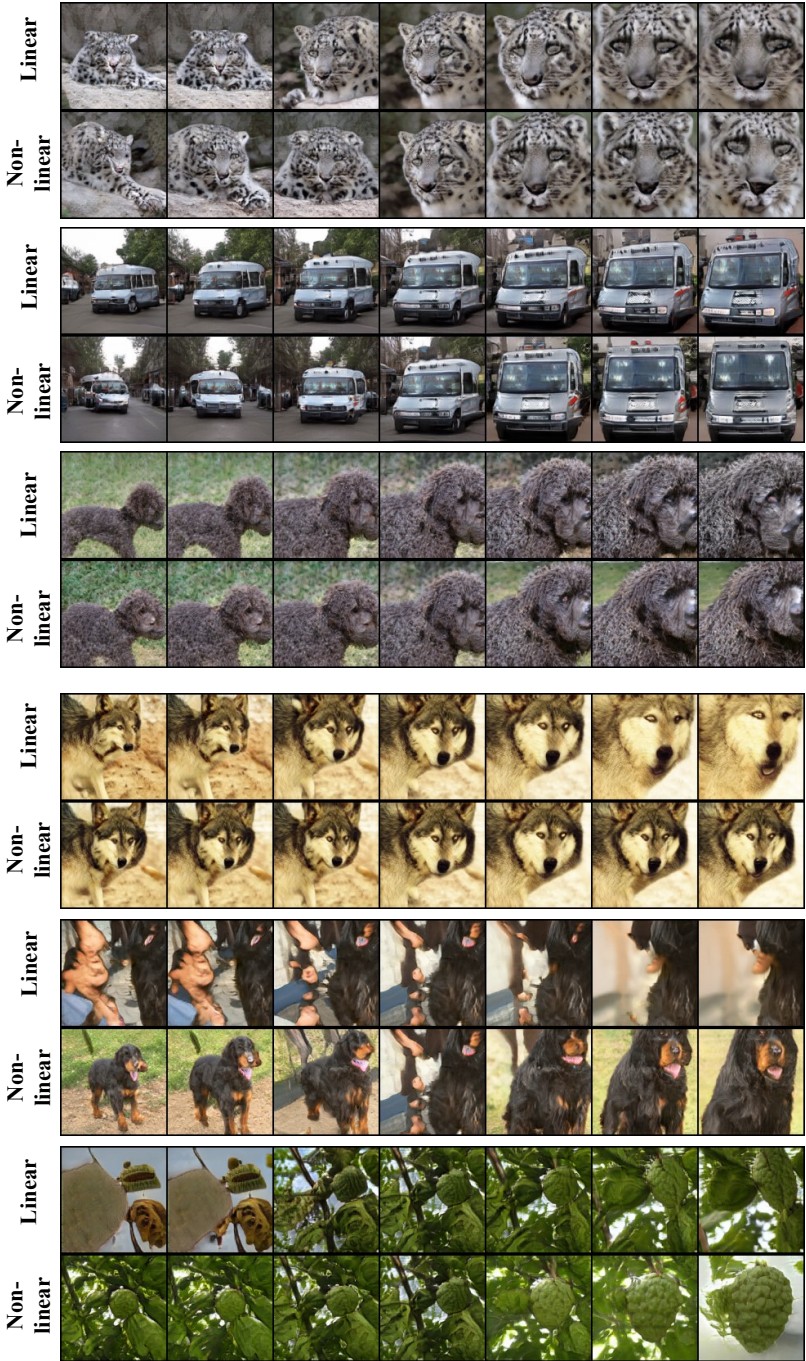

Figure 51: User prescribed zoom with BigGAN. Our method, linear vs non-linear trajectories

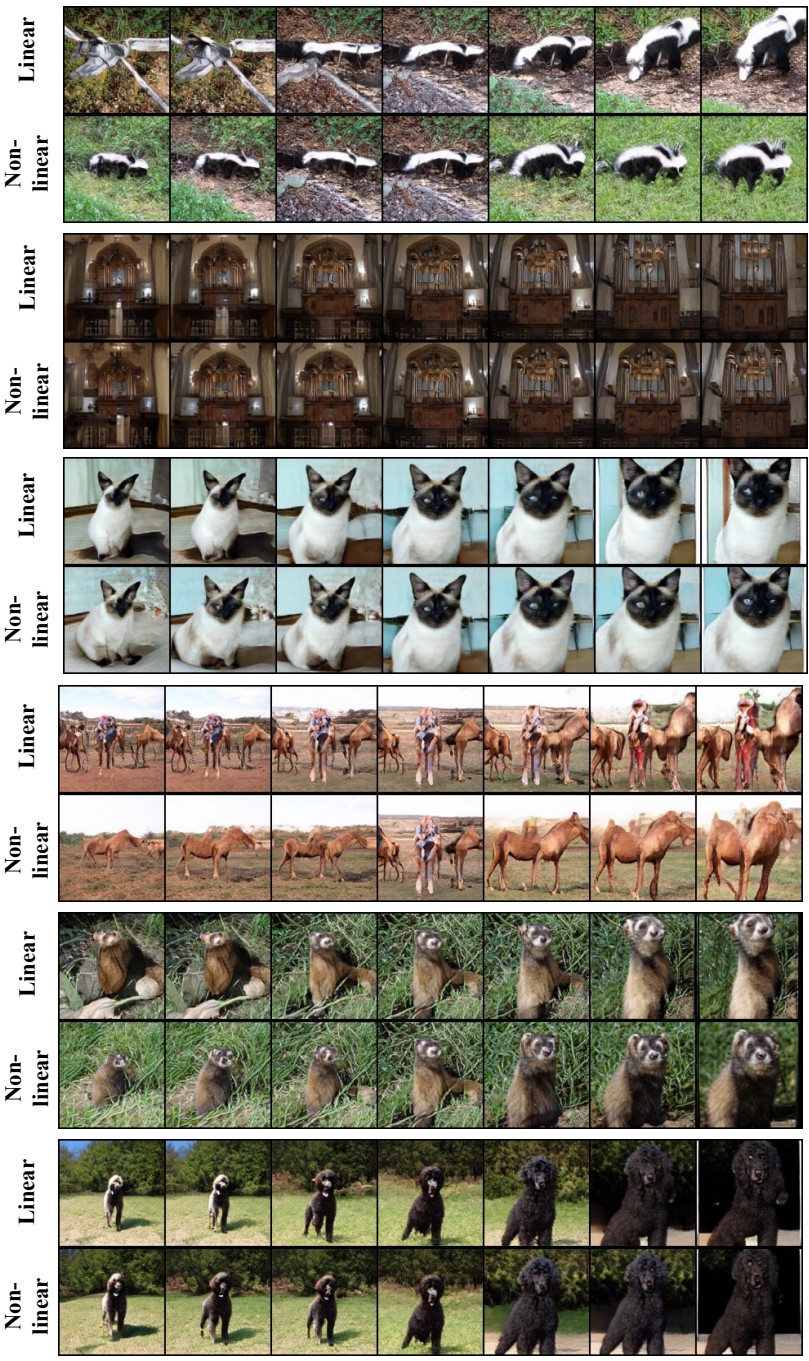

Figure 52: User prescribed zoom with BigGAN . Our method, linear vs. non-linear trajectories.

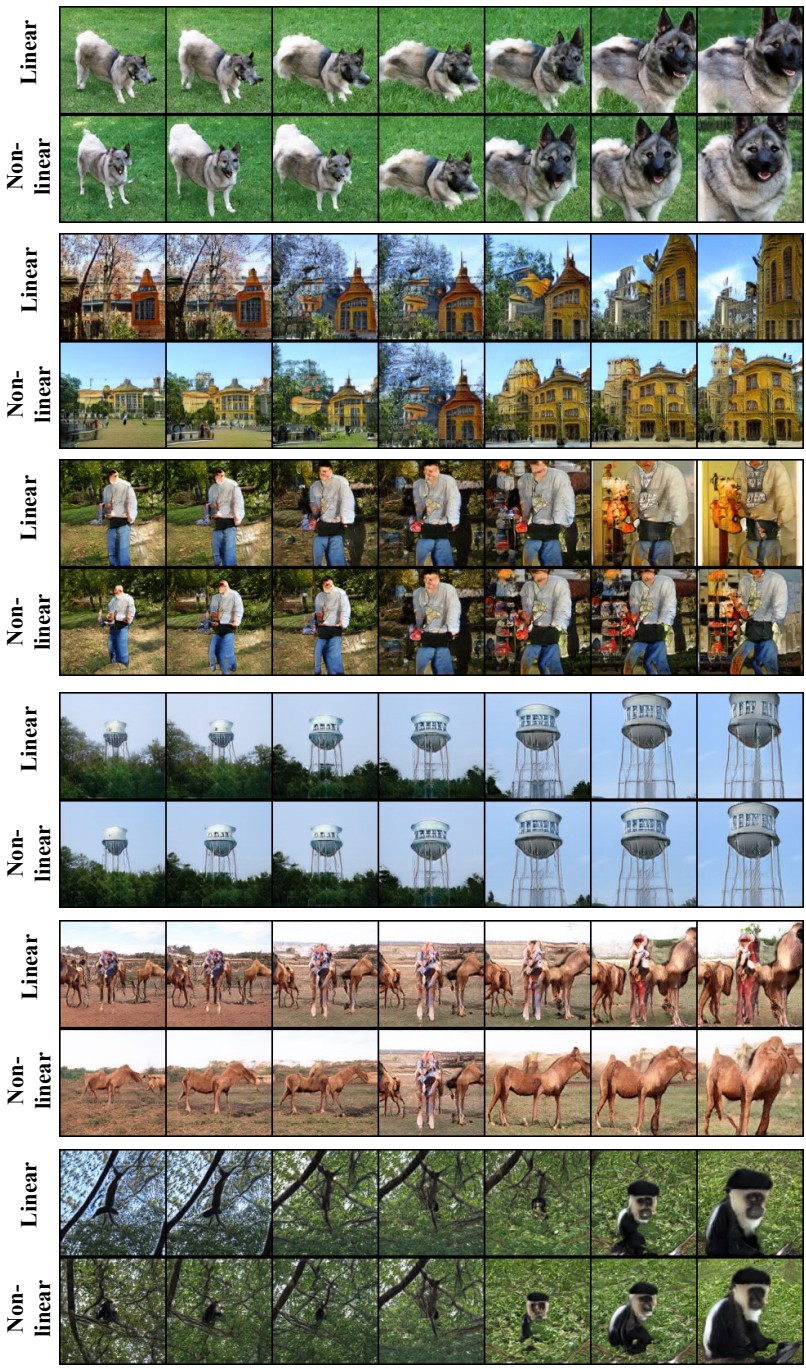

Figure 53: User prescribed zoom with BigGAN. Our method, linear vs. non-linear trajectories.

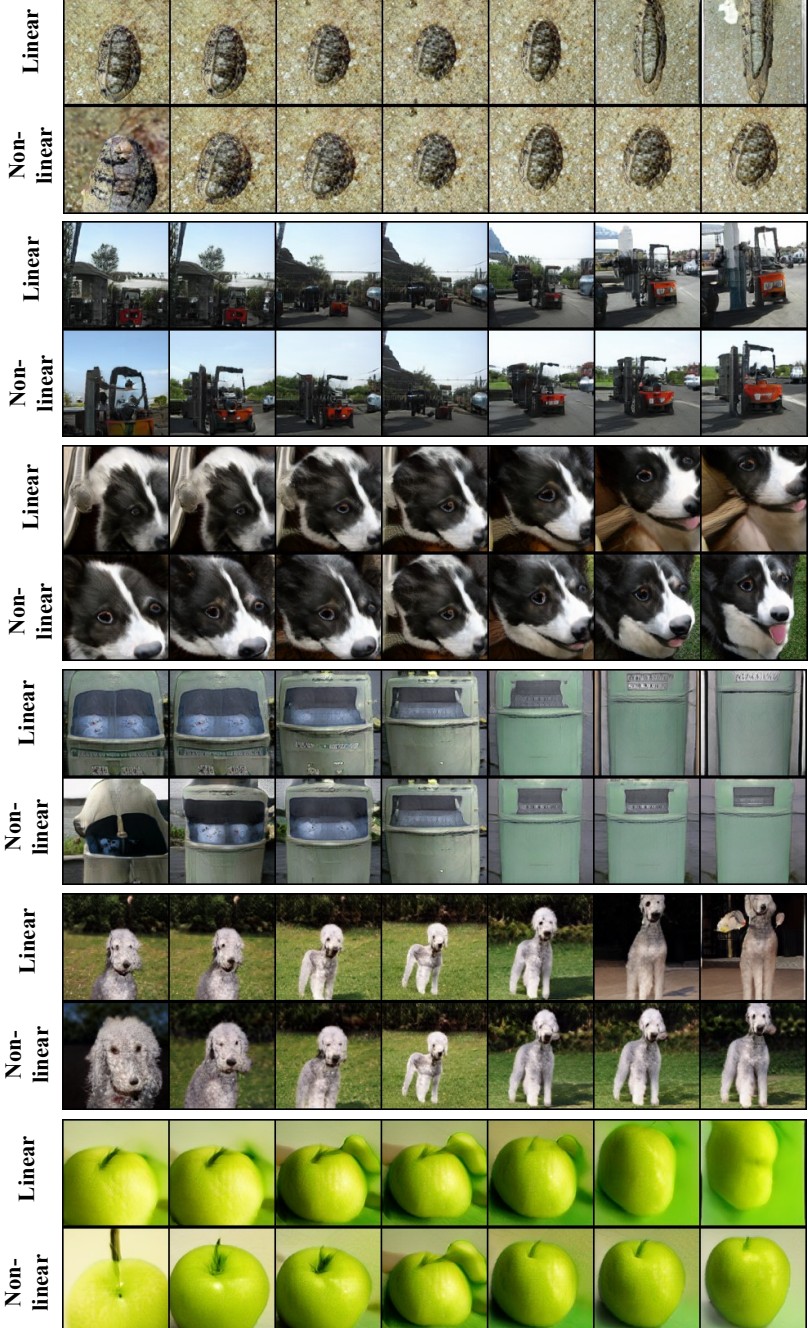

Figure 54: User prescribed vertical shift with BigGAN. Our method, linear vs. non-linear trajectories.

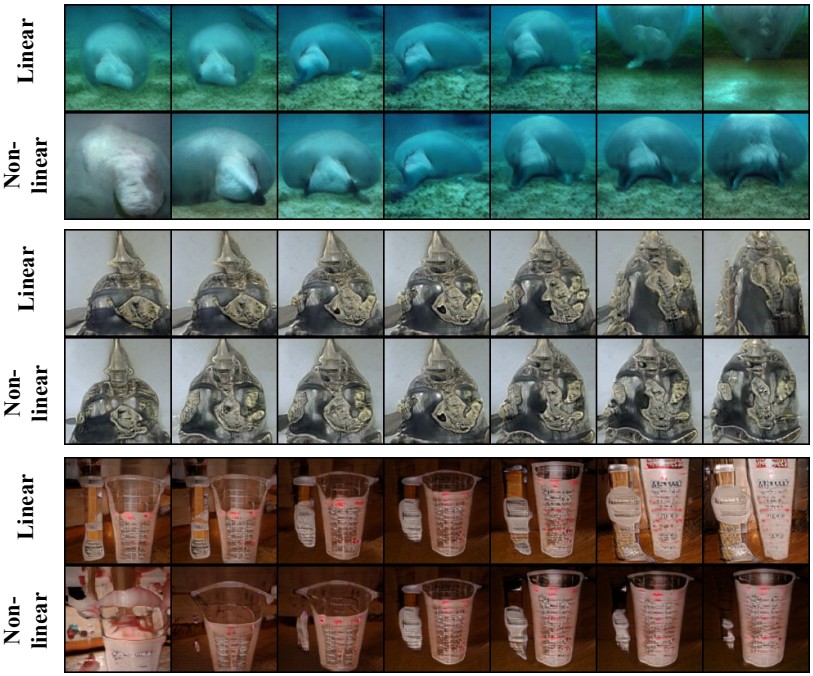

Figure 55: User prescribed vertical shift with BigGAN. Our method, linear vs. non-linear trajectories.

**- Zoom +**    **- Zoom +**

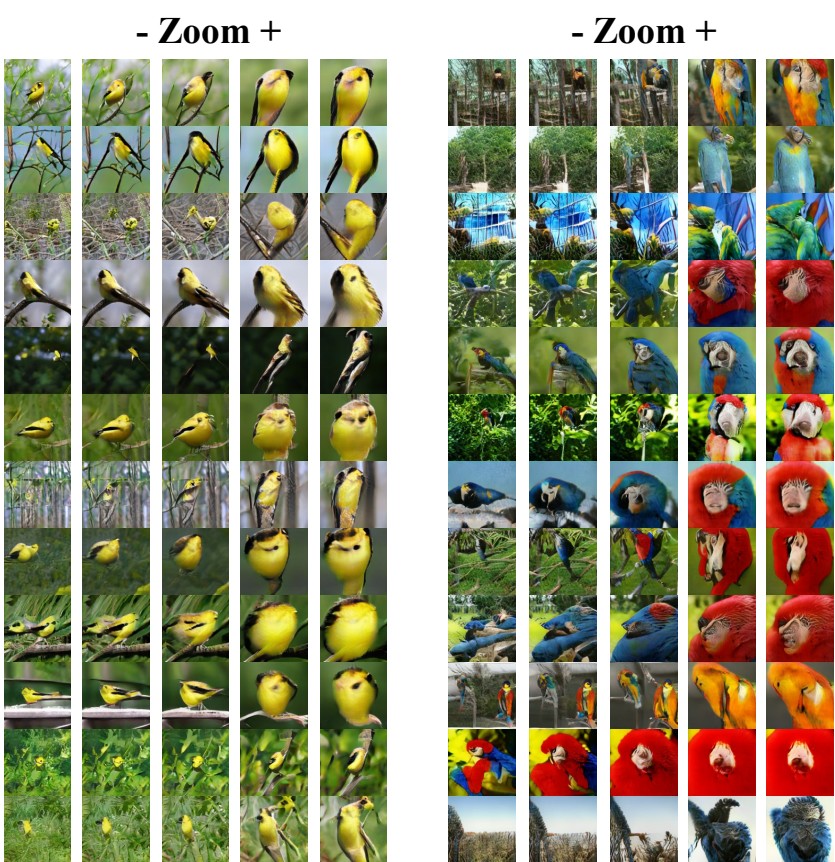

Figure 56: Zoom transformation with DCGAN.

**- Shift X +**          **- Shift X +**

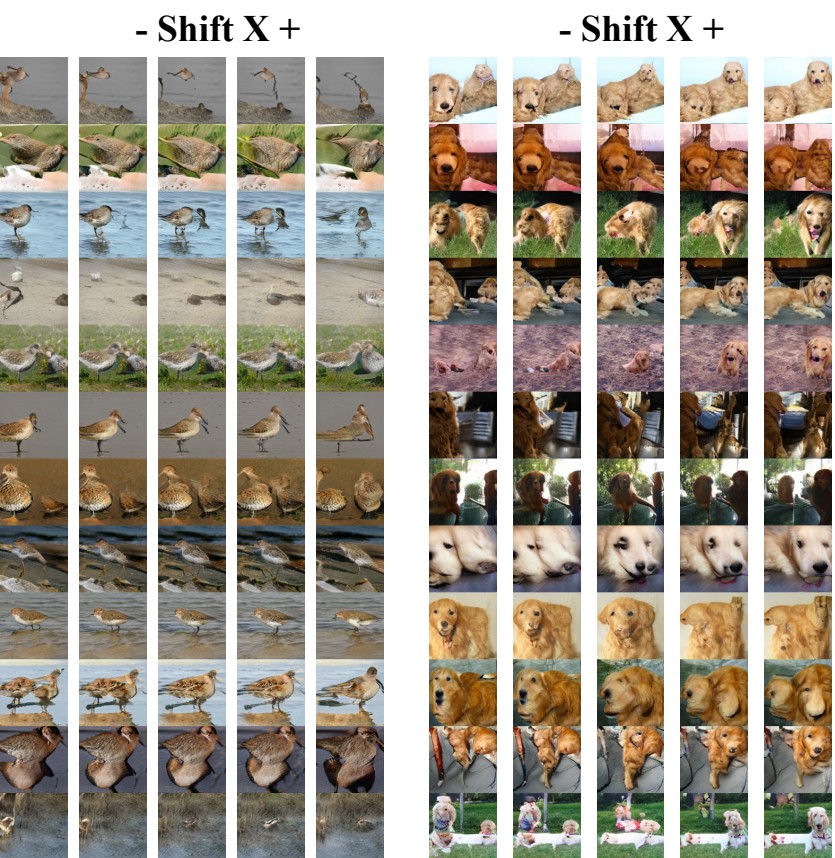

Figure 57: Shift X transformation with DCGAN.

## A.4 ATTRIBUTE TRANSFER

We next provide more attribute transfer examples. Figures 59,60 show pose transfer examples, which are obtained by swapping the part of the latent vector corresponding to scale 1. Figure 61 depict texture transfer examples, which correspond to swapping the parts of the latent vector and the class, corresponding to scales 3,4,5.

## - Shift Y +        - Shift Y +

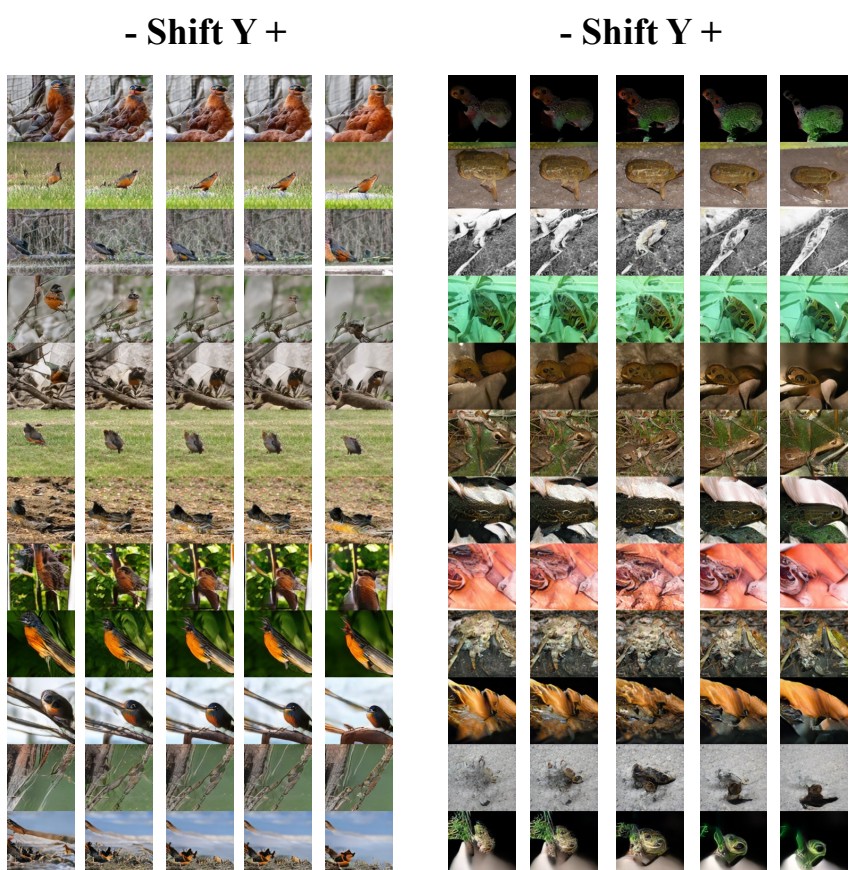

Figure 58: Shift Y transformation with DCGAN.

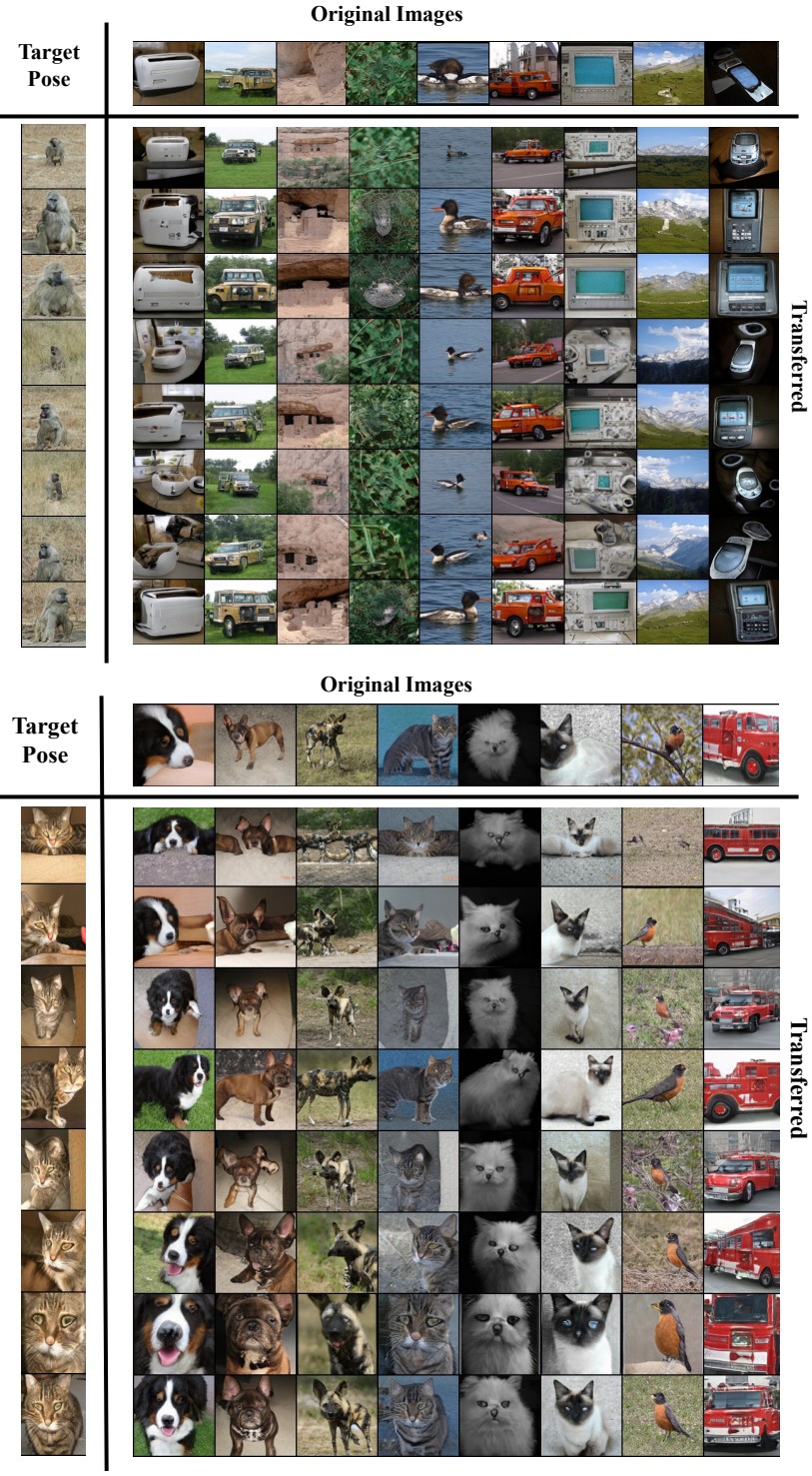

Figure 59: Pose transfer by swapping scale 1 of the latent vector.

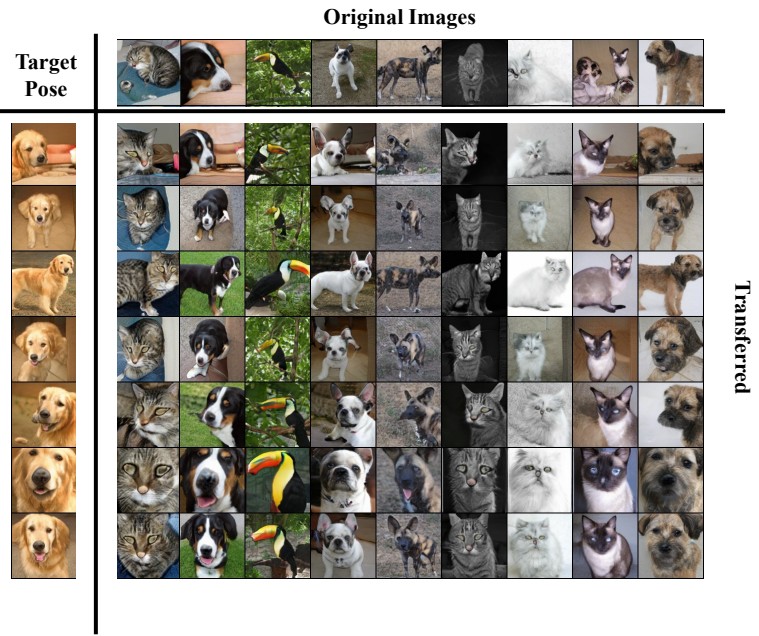

Figure 60: Pose transfer by swapping scale 1 of the latent vector.

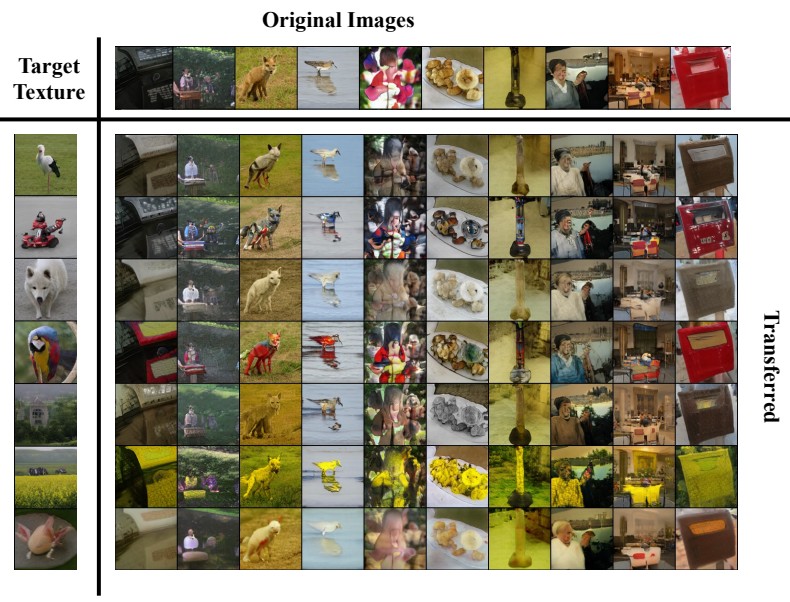

Figure 61: Texture transfer by swapping scales 3,4,5 of the latent vector.

