# OpenReview forum: "GAN "Steerability" without optimization "
_ICLR.cc/2021/Conference — ICLR 2021 Spotlight_

### Official Review · AnonReviewer2 · 2020-10-27
**Strong submission with clear novelties and comprehensive experiments**

**Rating:** 8
**Confidence:** 4

**Review:**

I am a picky reviewer in most cases. This submission is one of the highest-quality manuscripts I have reviewed. The problem setup is clear, the survey is comprehensive, and the contrastive to the prior arts is clear. I am pretty enjoyed reading this paper.

The task of finding meaningful steerability within the latent space is relatively a new research area. The authors have made a fairly complete review and investigation on the related works, then propose an unsupervised (meanwhile, still providing slight user controllability with a small set of transformations) and training-free algorithm with high-diversity and high-quality (in terms of the robustness of the trajectory found by the algorithm). Besides, the authors propose the concept of the first-order and second-order dataset biases, and tackles the problem with non-linear trajectories discovery. Also, highlight that the proposed method is significantly faster than the prior arts as shown in Table 1.

Experiment-wise, the authors provide a huge set of qualitative results in the appendix. Though the results are already convincing enough, I would still recommend the authors to provide some quantitative numbers for the comparisons on the diversity of latent directions against other methods (Figure 34-38). The visual differences are obvious, but a quantitative number can further showcase the universality of higher diversity. Some simple measurements like a mean over the variance of LPIPS among all dimensions [1] may be fair enough.

Despite the submission is pretty strong and I pretty much has no problem with it, I ended up rating it with an 8, as the impact is relatively narrow in a specific area/task. But I am willing to see the feedback from other reviewers and give it a certain level of adjustment.


[1] To avoid ambiguity, I mean
```
all_vars = []
for trajectory in all_trajectories:
	img_pairs = # interpolate the trajectory
	cur_var = np.var([lpips(*img_pair) for img_pairs in latent_pairs])
	all_vars.append(cur_var)
mean_var = np.mean(all_vars)
```
This is just a demonstration, any implementation with a similar concept is fine.


**[Minor comments]**
1. While mentioning other related methods in the paper (e.g., Table 1), the references are made with plain text (without hyperlinks) and mostly using the authors' names in replacement of the method names. It is quite hard to keep tracking which reference corresponds to which method. It would be great if the authors can investigate this problem a bit.

2. Typo: Frobenious => Frobenius

**[Discussion]**
I do not count this section as a part of the review, so it wouldn't affect my scoring if the authors do not reply.
What is the main application of the dataset summarization technique (let's exclude image editing and content creation, which can be achieved with other techniques)? It is interesting to visualize these properties of the dataset, however, the real-world application of such a technique is not obvious and carefully investigated.
A quick answer might be finding the dataset bias or domain shifting, and we may explicitly fix it when these problems are revealed by these dataset summarization techniques. However, the works on dataset summarization have never explicitly shown if these applications are viable with some experimental setup.
I am interested in what is the authors' perspective on the applications, and whether showing these applications is an important part of the dataset summarization research direction, as the applications often influence the impact and research direction of proposed new settings and new components.

---

> ### Author Response · Authors · 2020-11-21
> **We thank you for the great review and wonderful suggestions and insights. We’re glad you enjoyed the paper and found it interesting. Below we address each of your comments.**
>
> **Quantitative evaluation of the diversity of latent directions:** Excellent suggestion. We added the LPIPS distance averaged over all first 50 directions. Specifically, we measured the LPIPS distance between pairs: [original image, 3 steps to left] and [original_image, 3 steps to the right]. We added those numbers to Sec. 3.1. For the first 25 directions, we got 0.036 and 0.059 for GANSpace and our approaches, respectively. Therefore, our approach indeed leads to larger semantic dissimilarity for the same step size.
>
> **Minor comments:**
> - Unfortunately, other than GANSpace, the methods of Tab. 1 don’t have names. We therefore left them as citations, but made them into hyperlinks as you suggested. Other than the table, in the updated version we tried to use “GAN Steerability” and “GANSpace” as frequently as possible instead of plain references, for the works of (Jahanian et al.) and GANSpace.
> - Replace Frobenious with Frobenius: Fixed. Thanks.
>
> **Discussion:** This is actually quite interesting. We weren’t thinking of this visualization method in any context other than for understanding entanglement between attributes in generated GAN images. But this method can probably be used in any setting where visualizing coupling between attributes is important. Disclaimer: our research expertise is more in image generation and manipulation. But perhaps understanding such couplings in a training set can help design better classifiers, e.g. in transfer learning or domain adaptation. If in a certain dataset we have strong coupling between two attributes (say size of dog and location within the image), while we don’t expect such coupling to be strong in the target domain, then perhaps we can take this into account in the classifier design, or in the training images we choose to present to the classifier during training.

---

### Official Review · AnonReviewer3 · 2020-10-28
**Paper review**

**Rating:** 6
**Confidence:** 5

**Review:**

The authors propose two new techniques that extract interpretable directions from latent spaces of pretrained GAN generators. Both techniques are very efficient and are shown to work with the state-of-the-art BigGAN models. Furthermore, the authors describe additional details of the method, like determining the transformation end-points, which are important for usage in the practical visual editing.

Strengths:

1. The paper tackles the important problem, provides a thorough description of the field, demonstrates an in-depth understanding of the area. The important small details (determining the end-points, entangled transformations) are only slightly addressed in the existing literature, but are crucial for editing applications. This paper convincingly addresses this gap.

2. The proposed techniques are both simple and efficient, much faster compared to existing methods.

Weaknesses:

1. What upset me most was a certain amount of overclaiming about the main contributions.

"Second, it detects many more semantic directions than other methods." - I cannot agree with this statement.

a) For user-specified transformations, the method finds the same directions as  Jahanian et al. (2020).

b) For unsupervisedly discovered transformations, I did not find in the text any examples of directions, that are not covered by the:

Voynov & Babenko, 2020

Ha ̈rko ̈nen et al. (2020), 2020,

The Hessian Penalty: A Weak Prior for Unsupervised Disentanglement, ECCV 2020 (which is a missing related work, btw)

In my opinion, the statement is misleading, which is not acceptable when listing contributions.

2. I am not convinced by the significance of the search of user-specified geometric transformations, which take the largest part of the submission. Simple transformations, like zoom/shift/brightness, can be easily obtained automatically in editing applications, do we really need GANs for them?

3. "[Ha ̈rko ̈nen et al. (2020)] obtain a set of non-orthogonal latent-space directions that correspond to repeated effects. In contrast, our directions are orthogonal by construction, and therefore capture a super-set of the effects found by Ha ̈rko ̈nen et al. (2020)"

This sound like a bold claim. First, directions from Ha ̈rko ̈nen et al. (2020), while not being orthogonal by construction, de-facto still can be close to orthogonal. Second, from both the main text and appendix, I did not understand, why the authors' method captures a super-set of directions, no quantitative comparison is provided.

To sum up, my current evaluation is (5). The proposed method is obviously more efficient, but I do not consider this advantage as a very important one, since transformation search is performed only once. Given the weaknesses listed above, I cannot recommend acceptance.

AFTER REBUTTAL:

The authors have toned down some of their claims and I am increasing my score accordingly.

---

> ### Author Response · Authors · 2020-11-21
> **We thank you for your review and constructive comments. Here we address each of your concern:**
>
> **Detecting many more semantic directions is an overclaim:** We’re sorry we gave the impression of trying to overclaim. Although we’re certain we do capture more directions than the others (see detailed list below), proving this quantitatively is very hard. For example, although it’s obvious from Figs. 45-50 that many of the GANSpace directions are repeated, numerically proving that a certain specific semantic direction doesn’t exist among those 128 axes is difficult. We agree with you that without having an undeniable quantitative proof, this may be problematic to state the way we wrote it. Therefore, we slightly rephrased (changed to “seems to detect more semantic directions”), and added some indirect quantitative evaluations (see below).
> Please note that our claim regarding more semantic directions referred only to the unsupervised setting. In this setting:
>
> - Voynov & Babenko (on BigGAN) seem to find only Zoom, Rotate, Brightness, Background blur (or clutter), Background removal, and Shift Y (see their Fig. 5). We find all these directions for BigGAN as well as also e.g. Lighting, BW, Temperature (3 directions - redness, yellowness, greeness).
>
> - Hessian penalty (on BigGAN) seems to find only Zoom, Rotate, Smoosh Nose (relevant only for dogs), BW. Again, we find in BigGAN also e.g. Shift Y, Lighting, Temperature (3 directions - redness, yellowness, greeness), Background removal, Background clutter. We added a citation to this work.
>
> - GANSpace (on BigGAN-deep) indeed seemingly detects many directions, and following your comment we rephrased our claim to make it weaker. However, note from Figs. 42-46 that for the same step-size, the effects they obtain are significantly smaller than ours, and in many of their directions there seems to be nearly no effect at all. To quantify this, we measured the average LPIPS distance between randomly generated initial images and images obtained after 3 steps along each of the directions (with the same step size for both methods). We found that our method achieves a 1.6x larger average semantic distance than GANSpace. We added this to Sec. 3.1. This means that even if some semantic direction does exist in their set (although we don’t manage to see it in Figs. 42-46), to obtain a similar effect to the corresponding direction of our method (e.g. same zoom level for the zoom direction), they need many more steps. This behavior does seem to emerge because of the fact that the directions in GANSpace are not orthogonal. We added Fig. 48 to show the degree of non-orthogonality of their directions.
>
>
>
> **Significance of the search of user prescribed geometric transformations.** Thanks for this important point. Please note that our key goal here is not necessarily to propose a new kind of editing algorithm. We are mostly interested in understanding what information is encoded in the weights of a pre-trained generator, and how exactly this information is encoded. We managed to show that simple geometric transformations are encoded only in the first layer. And not only that, we also found a closed form expression for extracting these directions from that layer. We believe that this observation on its own is very important, as it advances our understanding of how the generators of GANs actually work.
> Application-wise, we absolutely agree that zoom\shift and brightness can be easily manipulated through means other than GANs. However, even from a practical standpoint, a GAN does have certain advantages. For example, it completes details at the borders when shifting an object, it generates more fine details when zooming in, and generally it changes images in a semantic-aware manner. Take for example, the flower in Fig. 3. As the flower shifts up, the GAN completes the details at the bottom of the image. We hope this clarifies our view.

---

### Official Review · AnonReviewer4 · 2020-10-28
**A simple approach to find closed form expressions for latent space directions corresponding to prescribed geometric transformations such as a shift or a zoom is presented. Also extended to curved trajectories in the latent space. Both linear and nonlinear latent space walks as well as walks corresponding to principal directions of the first layer weight matrix are obtained. The paper also makes use of semantic directions such as background effects, color, and texture changes.**

**Rating:** 6
**Confidence:** 4

**Review:**

The paper is clearly written. Its significance lies in transferring both user-prescribed transformations and unsupervisingly discovered transformations over a pre-trained generator’s weights to the generator output.  Also, it discovers endpoints for latent space trajectories.

Pros:
One nice feature of the  proposed technique is that it works on a pre-trained generator, and no extra training or optimization is required, therefore, it is efficient.
Endpoints of the walks are investigated through analyzing the convergence of the walks.
Further demonstrations with walks on the great circle and small circle on the sphere through an initial point  and a  principal direction are presented that give insights into the transformations obtained in the generated images.

Cons:
- Results in the paper display artifacts (e.g Figure 3, Figure 6), therefore raises the question of carefully picked results being displayed, indeed it’s hard to predict the overall quality of generations.
- Section 3.4 where the attribute transfer is described is not clearly written. Reading this paper, one is not able to understand how background or texture  transfer is obtained through this work. I believe that part is based on a recent work by (Voynov, Babenko 2020). They showed a technique to discover meaningful latent directions in an unsupervised way, which included similar attribute transfers this paper has shown. I believe this paper relies on that work in finding the  directions that correspond to semantically meaningful attributes, or this point was not very clearly explained.  If that’s the case, the contribution of this paper mainly boils down to certain geometric transforms or principal directions.

The method of (Voynov and Babenko 2020) also operated on a pre-trained generator and obtained semantically meaningful directions in an unsupervised way through only optimizing two other simple networks to obtain meaningful directions. As far as I understand their optimization is not very demanding at all, therefore, is also efficient. Hence, the point made here in terms of ultimate efficiency with no training leads to a trade-off in not being able to discover a richer set of transformations.

---

> ### Author Response · Authors · 2020-11-21
> **Thanks for the important points you raise. Below we address each of your comments:**
>
> **Artifacts:** We’re not sure we understand which artifacts you’re referring to, but we’d like to stress that the quality of the generated images is the same as that of the base BigGAN model we use. Specifically, the initial image in the path is always randomly drawn from BigGAN, so that it’s quality has nothing to do with our method. And all other images along our circle walks (whether great or small) have the exact same likelihood as the initial image by construction. Namely, these images have the exact same probability to have been drawn by BigGAN as the initial image in the first place.
> Perhaps your concern is rooted in the fact that different papers use different BigGAN variants, and thus visual quality may vary across papers. But this is not related to the manipulation method that operates on top of the trained method. In our paper, we used two variants. Specifically, in Figs. 3 and 6 you referred to, we used BigGAN-128 and BigGAN-deep-512, respectively. We didn’t cherry-pick the results. Please see many more examples in the appendix, where we simply randomly drew images and performed walks.
>
> **Attribute transfer:** We’re sorry for the limited focus we put on this in the initial submission. We now made this a separate section and added Fig. 8 to visualize the approach and show more results. Regarding your concern, this part is not based on any previous paper. The idea is as follows. We found that the principal directions we obtain for different hierarchies in BigGAN, control different attributes. This general observation that different levels correspond to different attributes is not new; it was already mentioned in StyleGAN. However, our key observation was that swapping between corresponding parts of the latent code can transfer attributes. For example, to obtain pose transfer, we swap the first chunk (elements 0-20 of the latent code) with that of the target image which depicts the desired pose. This approach is very simple, yet extremely effective, and to the best of our knowledge has not been reported before. We therefore believe that readers interested in GAN manipulation should find it useful. We hope this addresses your concern.
>
> **The method of (Voynov and Babenko 2020):** Thanks for raising this point. Regarding speed, we ran their code on the same multi-GPU setup we used to run all other methods. Their optimization process took 12 hours and 20 minutes. This is while our method takes only 327ms. So similarly to the other works we mentioned, our method is ~5 orders of magnitude faster than this method. We added that to Tab. 1.
> As for the set of manipulations, we believe we do capture a richer set of transformations. These include several color manipulations, such as moving from color to black-and-white, a lighting (“shadowing”) transformation, and temperature transformations such as redness, yellowness and greenness. Please see Figs. 9-39 in the appendix.

---

### Official Review · AnonReviewer1 · 2020-10-29
**A fun paper that presents elegant new methods with the one drawback being that it is a bit of a hodgepodge of ideas**

**Rating:** 8
**Confidence:** 4

**Review:**


This paper studies transformations in GAN latent space that map to meaningful transformations in the generated data. The main contribution is to derive closed form methods for discovering latent transformations that correspond to 1) geometric changes and 2) changes that capture principle components of model variation. The paper also contributes new methods for nonlinear latent transformations, disentangled transformations, and an application to attribute transfer.

I really like this paper. The capabilities demonstrated here aren’t dramatically new — other methods can achieve similar effects — but this paper achieves these effects in a new way, which has its own advantages.

The positives I see are:
+ Simple and elegant alternative to prior work on finding latent transformations
+ Nice qualitative and quantitative results
+ Practical benefits including speed up, analytical transformation end points, and better disentanglement

My main criticisms are:
- toward the end of the paper it’s a bit of a hodgepodge of ideas
- the ideas and methods in Section 3 are mostly disjoint from those in Section 2
- the attribute transfer application especially feels tangential, and receives minimal analysis or evaluation
- the experiment in Figure 7 is not fully convincing

I think everything in this paper is interesting, but the different sections don’t fully cohere together. Section 2 is great and seems to tell a complete story, about closed form solutions to finding geometric transformations. Section 3 then diverges into a few different directions, which don’t directly build on each other. When the circular trajectories are introduced, I’m left wondering: why not use these in Section 2? For example, would it be possible to derive closed form solutions for great circles that map to target transformations $\mathbf{P}$?

I do like the story about first and second order biases, which somewhat connects Sections 2 and 3. However I would like to see more connections made at the methodological level, or a clear justification for why methods from Section 2 where not used in Section 3 and vice versa.

Aside from the coherence of the story, there are a few smaller things that could be improved:

In Figure 4, it would be nice to show the data distribution, or the distribution from the non-transformed GAN samples. Otherwise it’s hard to tell if the transformations had an effect.

In Section 3.1 I think an equation would help clarify things, e.g., write out the SVD and refer to the right singular vectors by an algebraic symbol. Perhaps also contrast this with the equation for PCA from Harkonen et al.

The experiment in Figure 7 is not convincing to me. From the plots it’s hard to tell if the proposed methods are actually incurring less shift. Maybe plotting the delta from the original distribution would make the plot clearer? Most of all, I think some numerical metric should be defined to quantify the second-order biases, rather than just requiring visual inspection of the plots. It is convincing that the FID improves for the proposed methods, but not that the level of disentanglement between zoom and shift improves.

Minor comments:
1. Page 1: “the precise same effect” — I think this is an overstatement. There are substantial differences between how a single direction affects different classes. For example see Fig 4 of Jahanian et al. 2020. I agree that it is interesting that the directions have _similar_ effects across classes.
2. In Eqn 3, what if the matrix is singular? Do you use the pseudoinverse?

---

> ### Author Response · Authors · 2020-11-21
> **Many thanks for the thorough review and important comments. We’re happy you liked the paper. Below we address each of you concerns:**
>
> **Different sections don’t fully cohere together:** Thanks for this important observation regarding the paper structure and the connections between sections 2 and 3. Following your comment we slightly modified the structure. Let us try to clarify. GAN steerability involves two elements: (i) finding meaningful latent space directions, and (ii) constructing plausible walks from such directions. We presented two strategies for (i) (sections 2 and 3 in the original submission), and three nonlinear strategies for (ii) (subsections within those sections).
> The nonlinear walk we introduced in Sec. 2 is inapplicable to the setting of Sec. 3 because it’s specifically designed for a user prescribed transformation P.
> The small circle walk of Sec. 3, is also practically inapplicable to the setting of Sec. 2. This is because it requires an additional reference direction, which we take as the one with minimal influence (smallest singular value). The framework of Sec. 2 doesn’t provide a way for determining such a non-influential direction.
> The only strategy that is applicable to both settings is the great circle walk we initially introduced in Sec. 3. This is because once a direction is determined (whether in a supervised manner or not), we can always walk on the sphere towards this direction.
> Therefore, following your remark, we moved the great circle discussion to Sec. 2 and added a clarification of this point. Now the great-circle is illustrated both for user-prescribed transformations (we added it to Figs. 3-5) and for the unsupervised directions (Figs. 6,7).
>
> **The attribute transfer application feels tangential and receives little analysis:** Regarding the limited focus on this topic in the original submission, we agree and apologize. Now that we have an additional page, we added illustrations and a schematic figure explaining this task (see Fig. 8). As for the relevance of this topic, although the technique we present here is somewhat different in flavor than the other methods we discuss, we believe it’s very much related to the general story of the paper. It presents another latent space manipulation method that applies to a pre-trained generator and requires no optimization. Although attribute transfer was studied by others, to the best of our knowledge it has nowhere been reported that attribute transfer can be achieved by simply swapping specific chunks of the latent code. We believe readers interested in GAN manipulation should find this interesting and useful, and we hope that now with the added details, the reviewer sees this as a more integral part of the story.
>
> **In Figure 4, show the distribution from the non-transformed GAN samples:** Good point, we added it.
>
> **In Section 3.1 an equation would help clarify things:** Thanks. We added an explicit mathematical explanation (including equations) within the text both for our method and for GANspace.
>
> **The experiment in Figure 7 is not convincing. Maybe plotting the delta from the original distribution would make the plot clearer:** Kindly note that those plots indeed depict the PDF of the delta shift between the initial image and the final image. As can be seen, the mean of the distribution of the delta shift is close to zero for the small circle walks, and close to 0.1 for the other methods. To avoid confusion of future readers, we added the term “delta shift” to the caption, and also specified the mean delta shift incurred by each of the methods, as well as the mean area (zoom effect) for each method. We hope the new caption makes this clearer.
>
> **“The precise same effect” is an overstatement:** We modified it to “similar effect”.
>
> **In Eqn 3, what if the matrix is singular?** This matrix is not singular for any reasonable generator architecture. For example, in BigGAN, W is 24,576x20 and so the matrix (DW)^T(DW), which we invert, is 20x20. To see why it’s rank is practically always 20, note that it’s rank actually equals the rank of DW (one way to look at it is through the SVD decomposition, where each singular value Si becomes (Si)^2 and therefore we remain with the same rank) . Now, W practically always has a full rank because otherwise it would mean that the generator uses less dimensions than the (already ridiculously low) number of 20 dimensions allotted to it. The matrix D zeros out some of the rows of W (say one fourth of the rows for translation), but the number of remaining rows is still overwhelmingly larger than 20, and so the rank is still 20.

---

### Author Response · Authors · 2020-11-21
**A new version of our paper**

We thank the reviewers for their effort and the great comments and suggestions. We uploaded a revised version of our paper with changes that are detailed at each of the responses below.
Thanks!

---

### Decision · Program_Chairs · 2021-01-07
**Final Decision**

**Decision:**

Accept (Spotlight)

**Comment:**

The paper gives an elegant and efficient closed form solution for steering directions in the latent space of a pretrained GAN to to produce transformations in the image domain such as scaling and rotation etc,  this also extended to attribute transfer. The new method leads to "speed up, analytical transformation end points, and better disentanglement"  w.r.t to competitive methods.  All reviewers agreed on the merits of this work, and the good qualitative and quantitative results . The rebuttal addressed reviewers questions and concerns regarding the structure of the paper and its coherence.  Accept